Resource

# Mass spectrometry–based proteomic exploration of diverse murine macrophage cellular models

Jack Gudgeon, Abeer Dannoura (ORCID), Ritika Chatterjee, Frances Sidgwick, Benjamin BA Raymond, Andrew M Frey, José Luis Marin-Rubio (ORCID), Matthias Trost (ORCID)

Immortalised cell lines that mimic their primary cell counterparts are fundamental to research, particularly when large cell numbers are required. Here, we report that immortalisation of bone marrow–derived macrophages (iBMDMs) using the J2 virus resulted in the loss of a protein of interest, MSR1, in WT cells by an unknown mechanism. This led us to perform an in-depth mass spectrometry–based proteomic characterisation of common murine macrophage cell lines (J774A.1, RAW264.7, and BMA3.1A7), in comparison with the iBMDMs, as well as primary BMDMs from both C57BL/6 and BALB/c mice. This analysis revealed striking differences in protein profiles associated with macrophage polarisation, phagocytosis, pathogen recognition, and interferon signalling. Among the cell lines, J774A.1 cells were the most similar to the gold standard primary BMDM model, whereas BMA3.1A7 cells were the least similar because of the reduction in abundance of several key proteins related closely to macrophage function. This comprehensive proteomic dataset offers valuable insights into the use and suitability of macrophage cell lines for cell signalling and inflammation research.

## Introduction

Macrophages were first described by Élie Metchnikoff who primarily noted their phagocytic (Greek: phagein—to eat; kytos—cell) ability (Gordon, 2016). As APCs belong to the innate immune system, they act as part of the body's first line of defence against invading pathogens or damaging self-molecules by recognition of pathogen-associated molecular patterns (PAMPs) or damage-associated molecular patterns (DAMPs), respectively. Macrophages are highly plastic in nature, existing as a range of tissue-resident cells and adopting various polarisation states in response to different activating stimuli. These activation or polarisation states hold distinct gene signatures and drive varying macrophage functions. In reality, the macrophage polarisation state exists on a complex spectrum; however, a more reductionist approach is often used to delineate

distinct phenotypes (Murray et al, 2014). This binary nomenclature consists of M1 (classically activated or pro-inflammatory phenotype) and M2 (alternatively activated or anti-inflammatory phenotype) macrophages. Classical activation is driven primarily by stimuli such as IFNs or bacterial components such as LPS. These macrophages are primed for the promotion of tissue inflammation and pathogen killing by phagocytosis (Pauwels et al, 2017). Although alternative activation is stimulated by cytokines such as IL-4, IL-10, and IL-13, alternatively activated macrophages are more closely linked to immunoregulation, wound healing, and tissue remodelling, yet still have high phagocytic capacity. It has been posited that a more standardised nomenclature should be used when reporting macrophage activation studies, specifying the stimulation used rather than the activation state, the definition of which may differ between research groups. This nomenclature takes the form of M(IL-4), M(LPS), or M(IFN-γ) to remove any ambiguity (Murray, 2017).

Various degrees of macrophage activation, coupled with different phenotypes of macrophage throughout the body, introduce complex heterogeneity and difficulty in comparing macrophage subsets. To simplify biochemical investigations of macrophages, common cell lines or mouse strains are often used. Common murine macrophage models are used in an effort to standardise investigations into innate immunity and inflammation. Primary cell models such as BMDMs from C57BL/6 mice are the gold standard as these cells are not transformed to allow indefinite culture and therefore better represent the in vivo macrophage phenotype (Assouvie et al, 2018).

However, immortal cell lines such as J774A.1, RAW264.7, and BMA3.1A7 are frequently used in cell signalling research, providing a cost-effective and ethically responsible alternative to in vivo or primary cell models. The use of cell lines in place of primary cells helps to address the "three Rs," principles originally outlined by Russel and Birch in 1960, which advocate the replacement, reduction, and refinement of animal models in research (Russell & Burch, 1960). Both J774A.1 and RAW264.7 cells were obtained from BALB/c mice (Ralph et al, 1976; Raschke et al, 1978), whereas BMA3.1A7 cells were obtained from C57BL/6 mice (Kovacsovics-Bankowski & Rock, 1994). However, comparisons may be hindered by the use of different strains, as C57BL/6 macrophages exhibit a

---

Biosciences Institute, Newcastle University, Newcastle upon Tyne, UK

Correspondence: matthias.trost@newcastle.ac.uk

greater propensity for classical activation, whereas BALB/c mice tend to acquire a more alternatively activated phenotype when challenged with LPS (Mills et al, 2000). Consequently, employing differing strains to investigate macrophage response to infection may result in conflicting evidence (Dill et al, 2015; Guo et al, 2015).

Furthermore, although cell lines offer an unlimited supply of biological material and are easier to use and manipulate, there is a caveat to their use: they must retain functional features close to those of their primary counterparts. As immortal cell lines are obtained through virus-mediated transformation or derived from tumours, they are susceptible to the accumulation of additional genetic alterations over successive passages. Consequently, these alterations can result in changes in phenotype and the loss of essential functions that are specific to macrophages (Kaur & Dufour, 2012). To ensure that experimental results can be reliably translated into real-life physiology or pathophysiology, it is crucial to characterise and validate cell lines against primary cells. One common method used to immortalise BMDMs derived from mice employs the J2 recombinant gamma-retrovirus. The J2 virus originates from the replication-defective 3611-Moloney murine sarcoma virus (Mo-MSV) and harbours viral *raf* (*v-raf*) and viral *myc* (*v-myc*) oncogenes. The introduction of these oncogenes into BMDMs induces indefinite proliferation without the need for growth factors (Blasi et al, 1985; De Nardo et al, 2018). Cell lines generated using the J2 virus have been characterised using multiple different approaches, confirming the presence of common macrophage markers (Blasi et al, 1985), antigen presentation capability (Kovacsovics-Bankowski & Rock, 1994), and ability to polarise towards different phenotypes (Banete et al, 2015; Spera et al, 2021). However, comprehensive proteomic studies have not been performed.

Overall, macrophage biology is incredibly complex as phenotypes can be affected by tissue residence, polarisation state, and genetic background. It is therefore imperative that these variables are considered when designing investigations and interpreting experimental results. Furthermore, after the generation of a new model cell line, such as the immortalised BMDMs (iBMDMs) generated here by J2 virus infection, it is vital that characterisation steps are carried out to assess their validity as a cellular model. To this end, we report the characterisation of the commonly used murine macrophage cell lines (RAW264.7, J774A.1, and BMA3.1A7), alongside WT iBMDMs in comparison with WT C57BL/6J and BALB/c BMDMs. To facilitate this, comprehensive liquid chromatography–tandem mass spectrometry (LC-MS/MS) data-independent acquisition (DIA) was used to interrogate the proteome of these cell lines.

## Results

### Expression of MSR1 is lost after immortalisation of BMDMs

This project intended to continue studies (Guo et al, 2019; Govaere et al, 2022) into the role of MSR1, the macrophage scavenger receptor 1. MSR1 has been shown to have activity in macrophage polarisation, pro-inflammatory signalling, pathogen clearance, Alzheimer's disease, atherosclerosis, non-alcoholic fatty liver disease, and cancer (Gudgeon et al, 2022). To reduce the number of animals needed for biochemical experiments, the project started by generating a cell line harbouring a genetic KO. Because standard CRISPR/Cas9 methods proved difficult because of *Msr1's* short exons, immortalised cell lines from WT and *Msr1* KO BMDMs were generated using the J2 recombinant gamma-retrovirus (De Nardo et al, 2018).

WT and *Msr1* KO iBMDMs were generated from C57BL/6J mice as described in the Materials and Methods section (Fig S1). Briefly, BMDMs were transduced with the J2 virus sourced from the AMJ2-C11 cell line. The concentration of L929 conditioned medium in the bone marrow growth medium was then gradually reduced to 0% over 2–3 mo. iBMDMs were confirmed to be macrophages by analysis of the cell surface receptors F4/80, CD11b, and CD11c. WT iBMDMs were confirmed to be F4/80$^{high}$CD11b$^{high}$CD11c$^{low}$ (Fig S2). The expression of CD11c, a marker for dendritic cells, was reduced in iBMDM cells compared with WT BMDMs. This was potentially a result of the cells no longer being cultured in L929 conditioned media, as previous studies have shown that the L929 conditioned media used to culture BMDMs can induce CD11c expression (Rice et al, 2020; Heap et al, 2021).

Initial flow cytometry experiments showed that MSR1 was not detectable in both WT and KO iBMDMs, whereas it was detectable in WT BMDMs. We also added other macrophage/monocyte cell lines such as J774A.1 and RAW264.7 cells, which had high Msr1 expression, and BMA3.1A7 cells, which showed lower cell surface abundance (Fig 1A). To determine whether this loss of expression was driven by protein degradation or by changes at the genetic or epigenetic level, reverse transcription–quantitative polymerase chain reaction (RT–qPCR) analysis was performed to investigate changes in *Msr1* mRNA levels between WT BMDMs and WT iBMDMs (Fig 1B). The loss was also confirmed by mass spectrometry alongside loss in a different iBMDM cell line generated from C57BL/6Ntac BMDMs (Fig 1C). Taken together, this confirmed the loss of *Msr1* gene expression seen after immortalisation, indicating that a possible epigenetic alteration, such as CpG methylation, or genetic alteration, such as gamma-retroviral integration into the genome, was responsible for the loss of MSR1.

### Proteomic characterisation of C57BL/6J iBMDMs and macrophage cell lines

Next, because of the unexpected loss of *Msr1* expression in iBMDMs, we aimed to assess the suitability of cell lines for studying innate immune responses compared with primary cells, employing a proteomic approach. Using a highly sensitive LC-MS/MS DIA method, we performed a detailed comparison between immortal macrophage cell lines (J774A.1, RAW264.6, BMA3.1A7, and iBMDMs) and primary WT C57BL/6J BMDMs. We identified 7,971 proteins using 2-h gradients on a Q Exactive HF mass spectrometer in a DIA mode, with 6,737 of the proteins being reliably identified through the detection of two or more peptides (Table S1). DIA-NN software was used to search against a sequence database in a library-free mode. Quality control checks showed uniform dispersion and intensity distribution in all four biological replicates and groups (Figs S3 and S4). Moreover, principal component analysis showed that the samples were separated from each other based on their protein composition (Fig S5). This suggests that there are significant

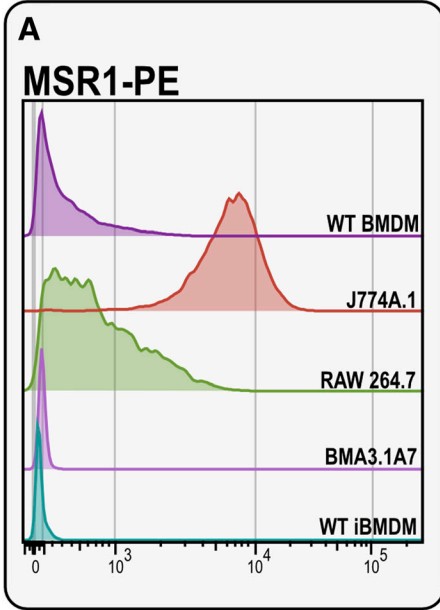
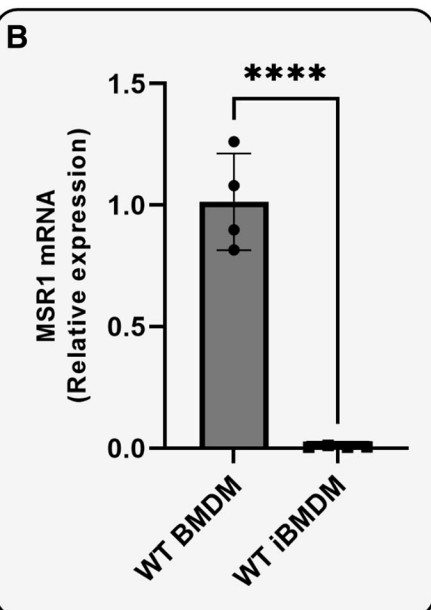
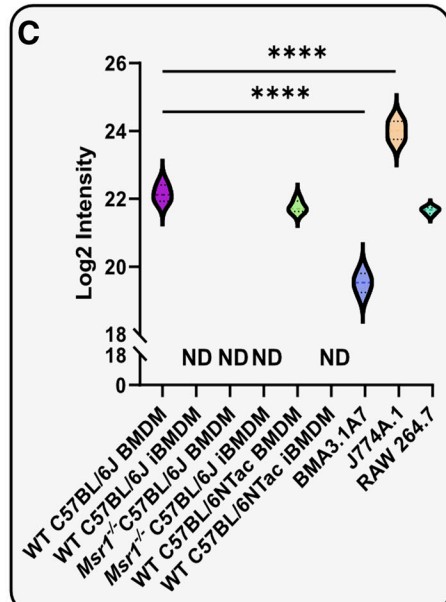

**Figure 1. Both gene expression and protein expression of MSR1 were lost after J2-mediated immortalisation of BMDMs.**
**(A)** Flow cytometry characterisation of macrophage cell lines. The surface expression level of MSR1-PE was compared between WT BMDMs, J774A.1, RAW264.7, BMA3.1A7, and WT iBMDMs. N = 1. **(B)** Quantification of mRNA levels of *Msr1* in WT BMDMs and WT iBMDMs (n = 4/group). Data are presented as the mean ± SD (unpaired *t* test); ****$P <$ 0.0001. **(C)** Proteomic log$_2$ intensity data of MSR1 across all cell lines and primary macrophages (N = 4). ND, not detected; ****$P < 0.0001$ by ordinary one-way ANOVA.

differences in the protein profiles of the cells in these different groups.

Although 16 peptides for MSR1 were detected overall in cell lines, MSR1 was not detected by mass spectrometry in WT iBMDMs, validating our flow cytometry data. Furthermore, J2-immortalised BMA3.1A7 cells displayed significantly lower MSR1 protein levels compared with WT BMDMs, whereas J774A.1 cells were shown to have significantly higher MSR1 levels than WT BMDMs. RAW264.7 cells showed similar protein levels compared with BMDMs, considering that BMDMs had lower MSR1 surface levels, this might imply that in BMDMs, there is a higher intracellular pool of MSR1 compared with RAW264.7 (Fig 1).

Overall, as anticipated, hierarchical clustering analysis revealed distinct clustering patterns between cell lines, with WT BMDMs forming a separate cluster from cell lines (Fig 2). Within this distinct cluster, J774A.1 cells exhibited the closest similarity to WT BMDMs, whereas BMA3.1A7 cells displayed the least similarity in terms of the proteome. RAW267.4 and iBMDMs cluster most closely together, likely because both were immortalised through the use of viruses. Furthermore, heatmap analysis was employed to depict relative protein abundance, which was divided into 12 distinct clusters based on hierarchical clustering of significantly different proteins.

The first cluster comprised MHC protein binding, TLR signalling, and endocytosis, which were increased in J774A.1, RAW 264.7, and BMA3.1A7 cells. The processes enriched in cluster 2 were predominantly associated with cellular homeostatic ion balance and were higher in all cell lines, except for BMA3.1A7 and WT BMDMs. Cluster 3 exhibited enrichment in all non-primary cell lines and was associated with terms related to the innate inflammatory response orchestrated by macrophages, including adenosine deaminase and

nitric oxide synthase regulator activity, and MHC binding. This is indicative of changes in different pathways that contribute to the eradication of pathogens. Regulation of retroviral genome regulation was observed, which may be lower in J774A.1 cells because of its origin from a murine cancer compared with the other cell lines, which were generated using retroviral methods.

Clusters 4, 5, and 6 were elevated in all cell lines. These terms linked to increased cell cycle activity, protein synthesis, metabolism, and biosynthetic process. These findings relate to the increased cell and protein turnover in non-primary cells, which occurs slower in primary cells as they reach senescence (Mathieson et al, 2018).

Cluster 7 referred to terms mainly suppressed in all cell lines. Suppression of podosome assembly, ERK1/2 cascade, regulation of early to late endosome transport, and defence response to bacteria indicates a decreased innate response ability of cell lines.

The terms enriched in cluster 8 were suppressed highly in J774A.1 and BMA3.1A7 cells, and all relate strongly to the uptake and clearance of particulate materials by macrophages. J774A.1 cells most closely matched BMDMs in cluster 9, which again link closely to phagocytosis.

The terms enriched in cluster 11 were down-regulated, specifically in BMA3.1A7 cells. The suppression of these terms indicates that BMA3.1A7 cells may possess altered inflammatory signalling and phagocytic capacity.

Finally, in cluster 12, J774A.1 cells were seen to have suppressed activity of the enriched processes. These include the pro-inflammatory stress–activated protein kinase (SAPK/JNK) signalling cascade and insulin-like growth factor receptor binding.

Overall, the comprehensive comparison between macrophage cell lines and primary macrophages presented in this study offers a

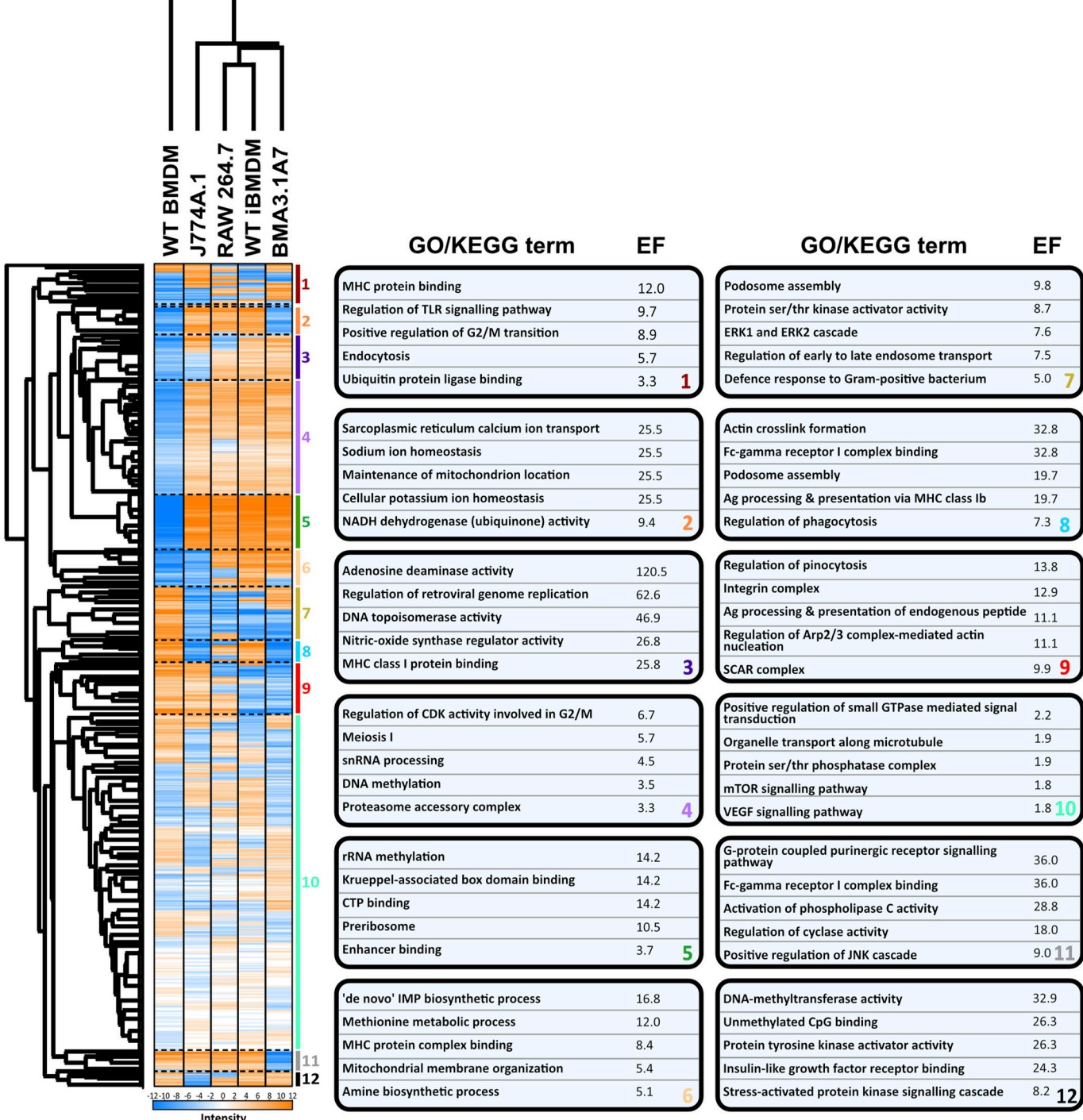

**Figure 2. Hierarchical heatmap analysis shows the variation between primary macrophage cells and cell lines, highlighting the enriched GO or KEGG terms in each specific cluster.**

Proteins were normalised by Z-score, filtered for at least three valid values in each group, and processed using ANOVA statistical testing with a Benjamini–Hochberg FDR correction cut-off of 0.05; ANOVA significant hits were then submitted to a post hoc test with an FDR of 0.05, and the heatmap was generated. Cluster analysis was performed to generate the enrichment factor (EF) of specific terms in 12 clusters using the Fisher exact test with a Benjamini–Hochberg FDR threshold of 0.02. All GO/KEGG terms displayed had a *P*-value < 0.0005. N = 4.

broad understanding of their similarities and differences. It sheds light on various crucial processes that may exhibit alterations across different cell types. However, given the extensive size of this dataset, more targeted pairwise comparisons were performed to identify and establish specific differences and proteins that could potentially account for these variations.

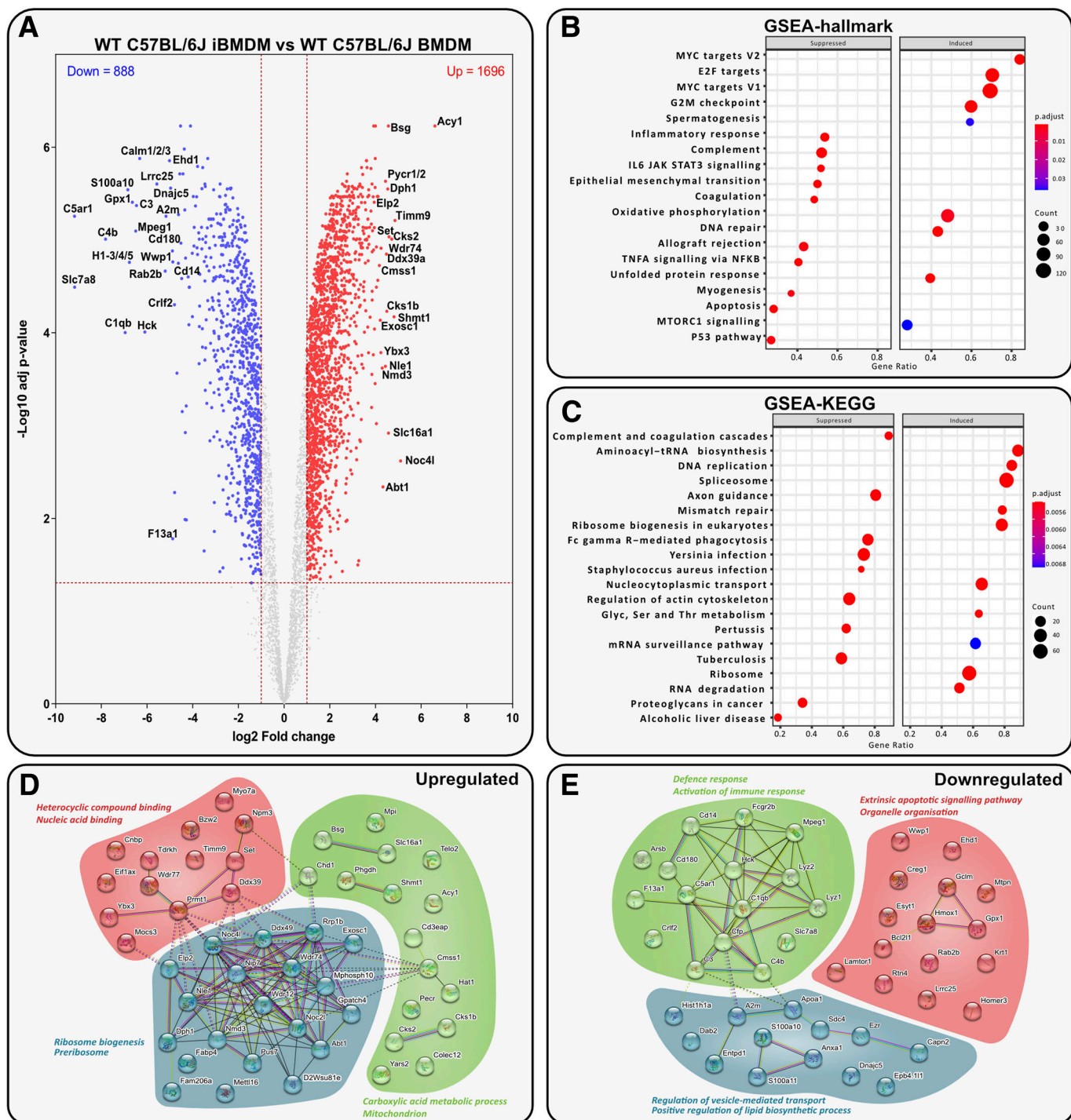

**Figure 3. J2-mediated immortalisation drives significant changes in the proteome of BMDMs.**
**(A)** Volcano plot displaying *t* test data of WT iBMDMs versus WT BMDMs (log$_2$ fold change > 1 or < −1; > 1.3 −log$_{10}$ [adjusted *P*-value]). Top 20 up- or down-regulated proteins are annotated. **(B, C)** Analysis of WT iBMDMs versus WT BMDMs shows the top suppressed or induced (B) hallmark gene sets defined by MSigDB and (C) gene sets defined by KEGG. **(D, E)** STRING analysis of the top 50 up- and down-regulated proteins in WT iBMDMs versus WT BMDMs.

## J2-mediated immortalisation drives significant changes in the inflammatory response and phagocytic activity of iBMDMs

Among the identified proteins, 49% show significant differential expression between primary WT BMDMs and WT iBMDMs, with 1,696

up-regulated and 888 down-regulated in WT iBMDMs (Fig 3A). Gene set enrichment analysis (GSEA) of the entire set of regulated proteins (Fig 3B and C) and STRING analysis (Fig 3D and E) of the top 50 differentially expressed proteins provided insights into altered protein expression patterns.

The analyses indicated down-regulation of the pro-inflammatory IL-6/JAK/STAT3 signalling axis and a broader inflammatory response in iBMDMs. Conversely, oxidative phosphorylation was induced, suggesting an anti-inflammatory profile compared with WT BMDMs. The complement and coagulation cascades were significantly suppressed, as were Fc-gamma–mediated phagocytosis and bacteriolytic enzymes Lyz1 and Lyz2. STRING analysis revealed up-regulation of ribosome biogenesis in iBMDMs, essential for sustained growth.

Overall, iBMDMs exhibited substantial proteomic changes compared with BMDMs, with down-regulation of key immune responses and polarisation towards an anti-inflammatory phenotype post-immortalisation.

### J774A.1 cells hold the most similar proteome compared with WT BMDMs

J774A.1 cells were revealed to be hierarchically closest to primary macrophages (Fig 2). A total of 2,465 proteins were shown to be differentially expressed compared with WT BMDMs in Fig 4A. However, despite these differences, J774A.1 cells were found to be similar to BMDMs, with few vital processes or pathways affected. Some key proteins were identified as significantly differentially expressed (Fig 4A). For example, several proteins pertaining to anti-inflammatory signalling were significantly down-regulated, including GPNMB, MRC1 (CD206), and the tetraspanin CD9. The down-regulation of these proteins may indicate a decreased propensity for alternative polarisation in J774A.1 cells; however, terms relating to anti-inflammatory processes were not identified in gene set analysis, suggesting that the number of anti-inflammatory proteins seen to change was not significant. Two of the main up-regulated proteins, aurora kinase B and pyrroline-5-carboxylate reductase 1/2 (PYCR1/2), are oncogenes with increased expression in multiple malignancies and associated with MYC in cancer and therefore link to the MYC target hallmarks shown in Fig 4B (Burke et al, 2020; Zhao et al, 2022).

Our results indicated that J774A.1 cells might have decreased pathogen-killing ability with KEGG terms such as *Salmonella* and *Yersinia* infection suppressed, alongside reduced endocytosis and lysosome activity (Fig 4C). This indicates that proteins involved in the binding of specific pathogens and subsequent downstream signalling are down-regulated. Among these, MRC1 is one such down-regulated protein, linked to the recognition and endocytosis of various microorganisms including *Yersinia pestis* (Azad et al, 2014). The main down-regulated interacting proteins were associated with S-100/ICaBP-type calcium binding domain and phospholipase A2 inhibitor activity (Fig 4E). In addition, the main up-regulated proteins in J774A.1 cells are interconnected (Fig 4D) and related to ribosomal and pre-ribosomal functions such as the Noc complex and Mpp10 complex, which are known to play significant roles in protein biosynthesis.

Overall, J774A.1 cells hold the closest proteome profile to BMDMs, as evidenced by the low number of macrophage function–related proteins showing significantly differential expression. However, changes related to anti-inflammatory pathways were seen, with decreased abundance of proteins such as GPNMB, CD206, and CD9.

Pro-inflammatory signalling via S100 proteins was also identified by STRING analysis as down-regulated. This again indicates altered inflammatory signalling in immortal cell lines.

### Inflammatory response is altered in RAW264.7 cells compared with WT BMDMs

RAW264.7 cells exhibit lower similarity to primary WT BMDMs compared with J774A.1 cells (Fig 2). Some proteins differentially expressed in J774A.1 cells, such as CD9 down-regulation and PYCR up-regulation, are also observed here, indicating cancer-driven metabolic reprogramming. Expression changes of proteins that directly relate to macrophage function were observed (Fig 5A). Down-regulation of pro-inflammatory protein MPEG1 and up-regulation of insulin-like growth factor 2 receptor (IGF2R) suggest decreased pro-inflammatory ability. However, up-regulation of GLUT1 may enable RAW264.7 cells to acquire a pro-inflammatory metabolic phenotype more readily. Increased pro-inflammatory activity is further supported by the up-regulation of cellular nucleic acid–binding protein and MHC class I member H2-D1. Pre-ribosomal proteins are also significantly up-regulated in RAW264.7 cells (Fig 5D).

GSEA reveals changes in the IL-6/JAK/STAT3 signalling axis and inflammatory response hallmarks (Fig 5B and C). KEGG terms related to infections and phagosome activity are down-regulated in RAW264.7 cells, suggesting a weakened ability to signal in a pro-inflammatory, host-protective manner. Similar to iBMDMs, complement and coagulation-related proteins (C3, A2m, C4b, C5ar1) were significantly down-regulated (Fig 5E).

### The BMA3.1A7 proteome profile is significantly different to that of WT BMDMs

BMA3.1A7 cells were identified as least similar to WT BMDMs by hierarchical clustering (Fig 2); this separation is recapitulated in Fig 6, which displays a wide range of important differences. Several proteins vital for normal macrophage function were identified as significantly down-regulated (Fig 6A). These include MPEG1, CD84, TLR-13, LYZ1/2, and CD206. These changes all indicate a decreased ability of BMA3.1A7 cells to recognise and respond to pathogens, overall resulting in lower pro-inflammatory M1 activity and signalling (Sintes et al, 2010; Kolter et al, 2016; Ragland & Criss, 2017; Abdelaziz et al, 2020; Bayly-Jones et al, 2020).

GSEA results show that the inflammatory and IFN-γ responses are suppressed (Fig 6B), as well as *Staphylococcus aureus* infection response (Fig 6C). Moreover, the down-regulation of TLR-13 may contribute to the decreased response to *S. aureus* infection, as it has been shown to be essential for its recognition in BMDMs (Kolter et al, 2016).

STRING analysis highlighted a highly interconnected cluster of up-regulated proteins, containing the highest number of proteins relating to the ribosomal and pre-ribosomal process compared with each other cell line (Fig 6D). Making up part of this cluster are the DEAD/DEAH box helicase proteins. The other main constituents of the large cluster are WD40 repeat (WDR)–containing proteins. The WDR domain is a highly abundant protein interaction domain involved in a variety of processes including the ubiquitin–proteasome system, cell cycle

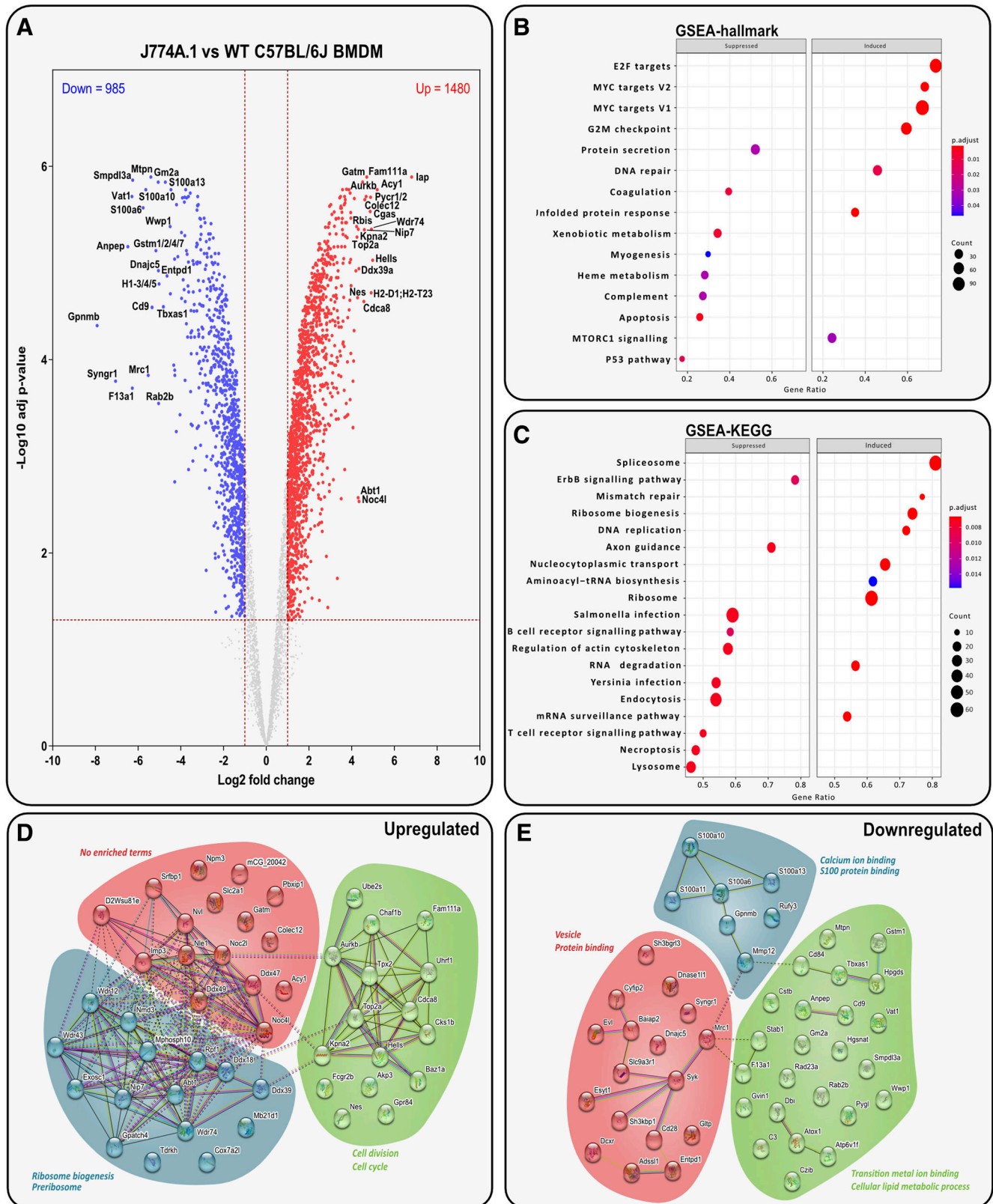

**Figure 4. J774A.1 cells hold the most similar proteome compared with WT BMDMs.**
**(A)** Volcano plot displaying adjusted $t$ test data of J774A.1 cells vs WT BMDMs ($\log_2$ fold change > 1 or < −1; > 1.3 −$\log_{10}$ [adjusted $P$-value]). Top 20 up- or down-regulated proteins are annotated. **(B, C)** Analysis of J774A.1 cells versus WT BMDMs shows the top suppressed or induced (B) hallmark gene sets defined by MSigDB and (C) gene sets defined by KEGG. **(D, E)** STRING analysis of the top 50 up- and down-regulated proteins in J774A.1 cells versus WT BMDMs.

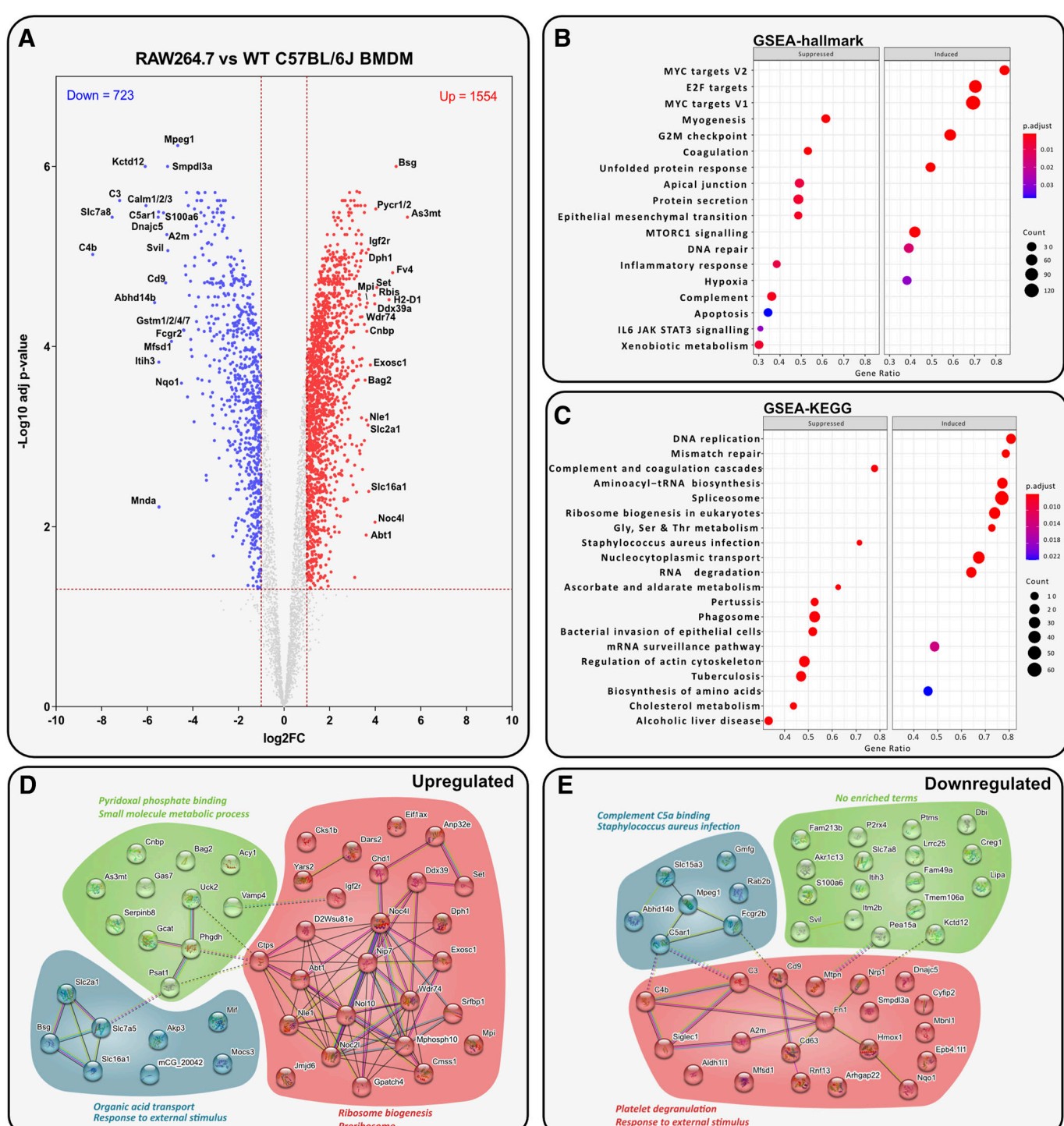

**Figure 5. Inflammatory response is altered in RAW264.7 cells compared with WT BMDMs.**
(A) Volcano plot displaying *t* test data of RAW264.7 cells versus WT BMDMs (log$_2$ fold change > 1 or < −1; > 1.3 −log$_{10}$ [adjusted *P*-value]). Top 20 up- or down-regulated proteins are annotated. **(B, C)** Analysis of RAW264.7 cells versus WT BMDMs shows the top suppressed or induced (B) hallmark gene sets defined by MSigDB and (C) gene sets defined by KEGG. **(D, E)** STRING analysis of the top 50 up- and down-regulated proteins in RAW264.7 cells versus WT BMDMs.

control, regulation of gene expression, and immune responses (Schapira et al, 2017; Jain & Pandey, 2018).

The down-regulated proteins identified by STRING analysis again link strongly to macrophage function and phenotype

(Fig 6E). Galectin-3 (Lgals3) and galectin-9 (Lgals9) are both linked to inflammatory signalling and damaged membranes. Proteins linked to the classical pathway of complement activation were also seen to be down-regulated again. Importantly,

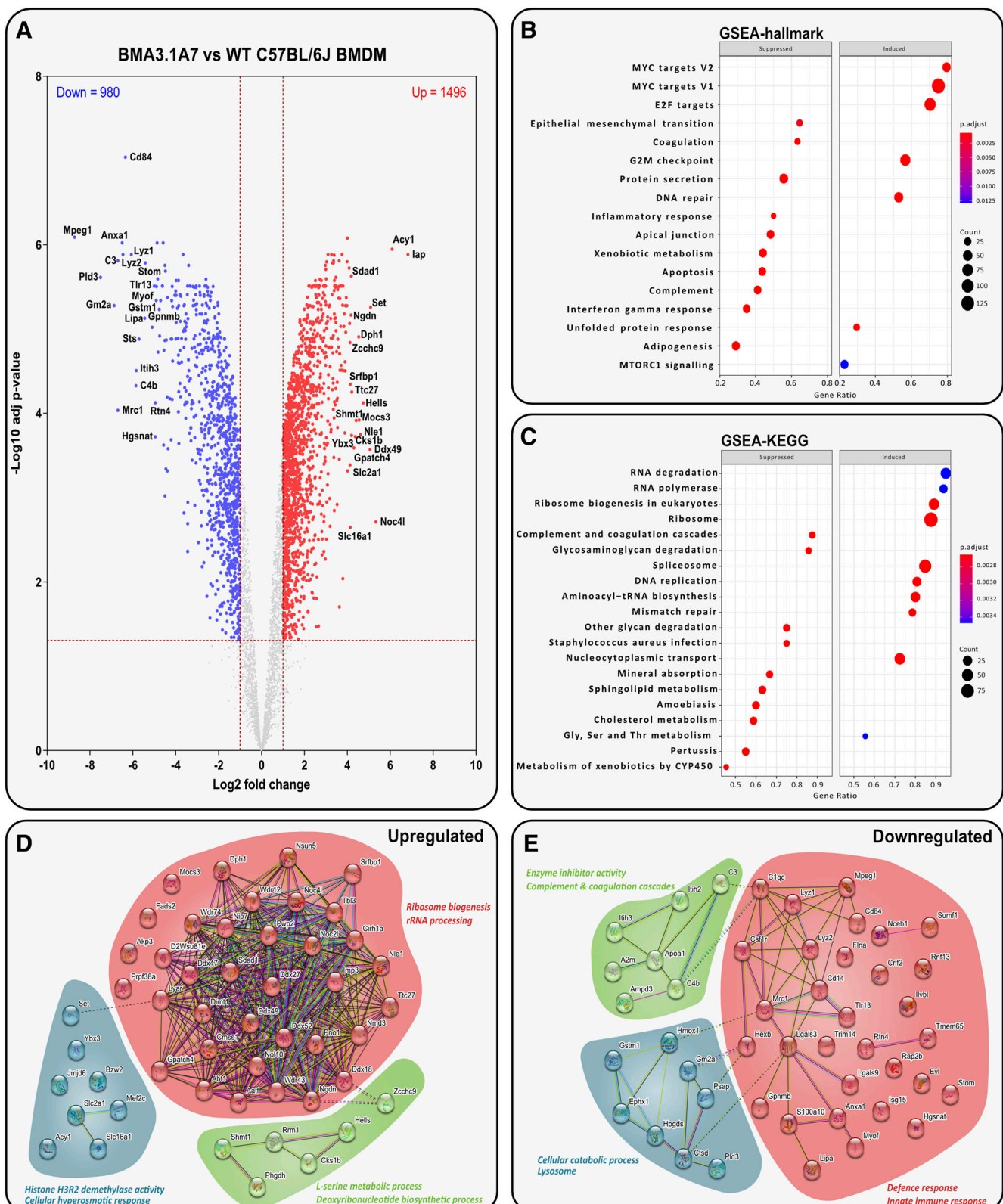

**Figure 6. BMA3.1A7 phenotype is significantly different to that of WT BMDMs.**
**(A)** Volcano plot displaying *t* test data of BMA3.1A7 cells versus WT BMDMs (log2 fold change > 1 or < −1; > 1.3 −log$_{10}$ [adjusted *P*-value]). Top 20 up- or down-regulated proteins are annotated. **(B, C)** Analysis of BMA3.1A7 cells versus WT BMDMs shows the top suppressed or induced (B) hallmark gene sets defined by MSigDB and (C) gene sets defined by KEGG. **(D, E)** STRING analysis of the top 50 up- and down-regulated proteins in BMA3.1A7 cells versus WT BMDMs.

BMA3.1A7 cells were the only cells to have the terms "BMDM" and "innate immunity" highlighted in the down-regulated STRING analysis.

Taken together, these results suggest that the BMA3.1A7 cell line exhibits the lowest degree of macrophage-like characteristics, as evidenced by the enriched terms identified in the down-regulated STRING analysis. Furthermore, most of the notable alterations observed were associated with significant changes in pro-inflammatory or pattern recognition receptor (PRR) signalling pathways, implying that this cell line may be less suitable for studies related to innate immune response.

### Macrophage receptor expression is significantly altered in macrophage cell lines

Macrophages employ a large catalogue of intracellular and surface receptors to act as sensors allowing identification of their surrounding cells, the state of the tissue, various metabolites, lipoproteins, antibodies, complement, and pathogens (Ley et al, 2016). These receptors act in tandem to regulate the protective function of macrophages. Therefore, it is essential that these receptors are expressed in cell lines at similar levels to primary macrophages. Macrophage receptors identified in this dataset were extracted along with their fold change in comparison with WT BMDMs (Table 1), providing potential insights into choosing the right cell line for experiments.

### Proteins crucial to macrophage function were unidentified in certain cell lines

Both global analysis and individual comparisons of proteome differences between cell lines and WT BMDMs gave valuable insights into the suitability of cell lines for the immunology and microbiology infection fields. However, these comparisons only considered proteins, which were identified in at least three of four biological replicates for each group. This meant that proteins completely unidentified in one cell type, but expressed in another, were not considered, as the data were not imputed. Such proteins might be of high importance when analysing experimental results or considering which cell line to use for specific investigations. We performed heatmap analysis of proteins identified in all four biological replicates of WT BMDMs but not in any replicate of at least one of the cell lines under investigation. These proteins were grouped into three hierarchical clusters labelled as (A), (B), and (C) (Fig 7).

Cluster (A) of the heatmap displays proteins that were mainly not detected in BMA3.1A7 cells. GSEA of this set of proteins revealed that in general, this cluster was related to the response to external stimulus, innate immune response–activating signalling pathway, and defence response (Fig 7A). In particular, scavenger receptor activity was highlighted in this cluster, with stabilin-1 (STAB1), CD36, and collectin-12 (COLEC12) all absent in BMA3.1A7 cells. Other proteins important for host defence were also not identified, such as transglutaminase 2 (TGM2), triggering receptor expressed on myeloid cells 2 (TREM2), and TLR-4.

Unlike in cluster (A), the proteins displayed in cluster (B) were not identified in various cell lines (Fig 7B). However, the proteins here were again linked to immune and defence responses. Several proteins relating to FcγR-mediated phagocytosis were not identified in various cell lines, potentially giving rise to differences in the uptake of opsonised particles. Myristoylated alanine–rich c-kinase substrate (MARCKS) was absent in RAW264.7 cells as was MARCKS-like protein 1 (MARCKSL1) in WT iBMDMs. These proteins affect processes such as cytoskeletal rearrangement, vesicular trafficking, and phagocytosis (El Amri et al, 2018). AKT2, a serine/threonine kinase, was not identified in BMA3.1A7, RAW264.7, and J774A.1 cells. Deficiency of this kinase shifts polarisation towards an M2-like state, negatively regulates TLR-4 signalling, and impedes the uptake of opsonised beads (Vergadi et al, 2017). Inflammasome function may be affected in RAW264.7 and BMA3.1A7 cells by the lack of PYCARD, an adaptor protein essential for the formation of the AIM2 and NLRP3 inflammasome (Fang et al, 2019). CD93, not identified in RAW264.7 and J774A.1 cells, and CD48, not detected in all three common cell lines, might also impact host protection against bacteria (McArdel et al, 2016; Nativel et al, 2019). Although varied in their functions, the proteins identified in this cluster are strongly related to inflammatory signalling and pathogen clearance. This indicates that cell lines might display varying responses to certain stimuli depending on which signalling pathway members are present or absent.

Cluster (C) contains proteins undetected in most of the cell lines and relates to coagulation, defence response to another organism, wound healing, immune response, and regulation of cytokine production (Fig 7C). IGF2R was not identified in J774A.1 cells. This protein, known to regulate anti-inflammatory metabolic reprogramming, was previously shown to be significantly up-regulated in RAW264.7 cells (Fig 5). Therefore, anti-inflammatory polarisation and signalling may differ between these two cell lines. S100a9 normally functions in complex with S100a8 in response to infection by inducing pro-inflammatory cytokines, reactive oxygen species, and nitric oxide. It also mediates rearrangement of the cytoskeleton and is therefore implicated in phagocytosis (Wang et al, 2018). H2-AA and H2-AB1, both class II histocompatibility antigens, are important for the presentation of antigens to CD4[+] T cells by MHC class II (Stables et al, 2011). Therefore, part of the adaptive immune response is impaired in these cells. Finally, IFN-γ receptor 1 (IFNGR1) was not identified in any cell line. IRNGR2, the second part of the IFNGR complex, was not identified in the mass spectrometry analysis. Because of the importance of the IFNGR receptor complex for macrophage function, this finding was further validated by flow cytometry. This indicated that contrary to the proteomic data, both IFNGR1 (Fig S6A) and IFNGR2 (Fig S6B) are expressed on the surface of all macrophage models, showing that the absence in proteomic datasets does not necessarily mean the total loss of the protein, but just indicates that protein levels are below the detection limit.

### Abundance of proteins related to pattern recognition and TLR-4 response is altered in BALB/c BMDMs compared with C57BL/6J BMDMs

Because the RAW264.7 and J774A.1 cell lines were derived from BALB/c mice, we in addition compared the proteome of BMDMs

**Table 1.** Macrophage receptor and DNA sensor expression in macrophage cell lines in comparison with WT BMDMs.

| Gene name | Function | J774A.1 | RAW264.7 | BMA3.1A7 | iBMDM |
|---|---|---|---|---|---|
| **PtdSer recognition receptors** | | | | | |
| Mertk | Regulates cell survival, migration, differentiation, and efferocytosis. | NS | ● (red) | ● (red) | ● (red) |
| **NOD-like receptors** | | | | | |
| Pycard | Mediates apoptosis, pyroptosis, and inflammation. | ● (blue) | ● (red) | ● (red) | ● (blue) |
| **C-type lectins** | | | | | |
| Clec12a | Mediates tyrosine phosphorylation of target MAP kinases. | ● (blue) | ● (red) | ● (red) | ●●● (blue) |
| **Cytokine receptors** | | | | | |
| Ifngr1 | Antimicrobial, antiviral, and antitumour response. JAK/STAT signalling. | ● (red) | ● (red) | ● (red) | ● (red) |
| **Complement receptors** | | | | | |
| C5ar1 | Binds C5a and stimulates chemotaxis and superoxide anion production. | ●● (blue) | ●●● (blue) | ●● (blue) | ●●● (blue) |
| **TLRs** | | | | | |
| Tlr2 | Innate response to bacterial lipoproteins. NF-κB activation. | ●● (orange) | ●● (orange) | NS | ●● (orange) |
| Tlr4 | Innate immune response to bacterial LPS. NF-κB activation. | ● (blue) | ● (blue) | ● (red) | ● (blue) |
| Tlr7 | Innate and adaptive response to ssRNA virus. NF-κB and IRF7 activation. | NS | ● (blue) | ●● (blue) | ● (blue) |
| Tlr8 | Controls response to pathogens via recognition of RNA degradation products. NF-κB and IRF7 activation. | NS | ● (red) | ● (red) | ● (red) |
| **Scavenger receptors** | | | | | |
| Scarb2 | Lysosomal receptor for glucosylceramidase (GBA) targeting. | ●● (blue) | ●● (blue) | ●● (blue) | ●● (blue) |
| Msr1 | Endocytic/phagocytic. Binds wide range of polyanionic ligands. | ● (orange) | ● (blue) | ●● (blue) | ● (red) |
| Cd68 | Phagocytic activity. Binds tissue- and organ-specific lectins and selectins. | ●● (blue) | ●● (blue) | ●● (blue) | ●● (blue) |
| Lgals3bp | Promotes integrin-mediated cell adhesion. | ● (blue) | ●● (blue) | ● (red) | ●● (blue) |
| Cd36 | Binds wide range of ligands. Angiogenesis, inflammatory response, fatty acid metabolism. | NA | ●● (blue) | ● (red) | ● (blue) |
| Scarb1 | Binds ligands such as phospholipids, lipoproteins, and phosphatidylserine. | NS | ● (blue) | ● (orange) | NS |
| Colec12 | Phagocytosis of Gram-positive and Gram-negative bacteria and yeast. | ●●● (orange) | ●● (orange) | ● (red) | ●● (orange) |
| **Fc receptors** | | | | | |
| Fcer1g | Adapter protein. Transduces activation signals from multiple immunoreceptors. | ● (blue) | ●● (blue) | ● (blue) | ●● (blue) |
| Fcgr1 | High-affinity receptor for the Fc region of IgG. | ● (orange) | ● (blue) | ● (blue) | ● (blue) |
| Fcgr2 | Low-affinity receptor for the Fc region of IgG. | NS | ● (red) | NA | ●●● (blue) |
| **Integrins** | | | | | |
| Itgam | Macrophage adhesion. Uptake of complement-coated particles and pathogens. | NS | ● (blue) | ●● (blue) | ●● (blue) |
| Itgb1 | Receptor for multiple ligands. Myoblast differentiation. | ● (orange) | ● (blue) | ● (blue) | NS |
| Itgb2 | Receptor for ICAM1-4 and ISG15. | ● (blue) | ● (blue) | ●● (blue) | ●● (blue) |
| Itgal | Receptor for ICAM1-4 and ISG15. | NS | ● (blue) | NS | NA |
| Itga4 | Receptor for fibronectin, MADCAM1, and VCAM1. | NS | ●● (orange) | ● (orange) | ●● (orange) |
| Itga5 | Receptor for fibronectin and fibrinogen. CD40 signalling. | ● (orange) | ●● (blue) | ● (blue) | ● (blue) |

| Gene name | Function | J774A.1 | RAW264.7 | BMA3.1A7 | iBMDM |
|---|---|---|---|---|---|
| Itga6 | Receptor for laminin. NRG1-ERBB, IGF1, and IGF2 signalling. | NS | ● (red) | ● (blue) | ●● (blue) |
| Itgb7 | Receptor for MADCAM1 and VCAM1. Adhesive interactions of leukocytes. | ● (red) | ●● (orange) | NS | ● (red) |
| Itgav | Recognises multiple ligands. NRG1-ERBB, IGF1, IGF2, and IL-1B signalling. | ● (blue) | ●● (blue) | ●● (blue) | ●● (blue) |
| Itgax | Receptor for fibrinogen. Monocyte adhesion and chemotaxis. | ● (red) | ● (blue) | ● (red) | NS |
| **DNA/RNA sensors** | | | | | |
| Adar | Edits both viral and cellular dsRNA. Can have proviral or antiviral effects. | ● (orange) | ● (orange) | ● (orange) | ● (orange) |
| Aim2 | Mediates inflammasome activation in response to dsDNA. | NA | NA | NA | NA |
| Cgas | Senses dsDNA, triggering type I IFN production. | ●●● (orange) | ●● (orange) | ●● (orange) | ●● (orange) |
| Ddx58 | Senses dsRNA, triggering type I IFN production. | NS | ● (orange) | ● (blue) | NS |
| Ifih1 | Senses dsRNA, triggering type I IFN production. | NS | NS | ●● (blue) | NA |
| Sting1 | Senses dsDNA, triggering type I IFN production. | ● (orange) | NS | ●● (orange) | ● (orange) |
| TLR3 | Senses dsRNA, triggering type I IFN production. | NA | ● (red) | ● (red) | ● (red) |
| TLR9 | Recognises unmethylated CpG dinucleotides. | NS | NS | NA | NS |
| ZBP1 | Recognises Z-RNA structures, mediating pyroptosis, necroptosis, and apoptosis. | NA | NA | ● (red) | ● (red) |

Protein name and functions are indicated. Fold changes are presented in coloured circles as indicated in the legend. Log$_2$ fold change (log$_2$FC) was calculated for each cell line in comparison with WT BMDMs. Unless otherwise stated, all values are significant at <0.05 adjusted *P*-value. N = 4.
● / ● = log$_2$FC of (±) 0.5–2; ●● / ●● = log$_2$FC of (±) 2–4; ●●● / ●●● = log$_2$FC of (±) >4; ● = not detected; NA = <3 valid values; NS = no significant change.

derived from BALB/c mice with the proteome of BMDMs derived from C57BL/6J mice in a separate mass spectrometry experiment. The same highly sensitive LC-MS/MS DIA method used in the cell line comparisons was used here. We identified 7,009 proteins, with 6,241 of the proteins being reliably identified through the detection of two or more peptides (Table S2). Quality control checks showed uniform dispersion and intensity distribution in all four biological replicates and groups (Figs S7 and S8). Moreover, principal component analysis showed that both samples were separated from each other based on their protein composition (Fig S9).

A total of 361 proteins differed significantly in their expression between BALB/c and C57BL/6J BMDMs. Of these, several pertain directly to macrophage function. CSF1, or M-CSF, along with its receptor CSF1R was significantly up-regulated, along with several PRRs, including COLEC12, CLEC6A, CXCR4, and MERTK. The hydrolytic enzymes LYZ1 and LYZ2 were also up-regulated. CD36, CD180, and LY86, all linked to TLR4-mediated LPS response were significantly down-regulated (Fig 8A).

Only four hallmarks were identified by GSEA: of note, oxidative phosphorylation was found to be induced in BALB/c BMDMs (Fig 8B). This was confirmed by STRING analysis, which highlighted a cluster of proteins pertaining to oxidative phosphorylation, as well as IgG binding, and bacteriolytic enzyme in the set of up-regulated proteins (Fig 8C). Clusters highlighted in the down-regulated set of proteins included amino acid biosynthesis, low-density lipoprotein (LDL) synthesis, and complement cascade regulation (Fig 8D).

## The phagocytic ability of J774A.1 cells is equal to that of primary BMDMs

As the expression of proteins relating to phagocytic processes was found to differ between macrophage cell lines and primary macrophages, we investigated the ability of different macrophage models to phagocytose bacteria (Fig 9A and B) and carboxylated beads (Fig 9C). In comparison with WT C57BL/6J and BALB/c BMDMs, and *Msr1–/–* BMDMs, J774A.1 cells were equally capable of phagocytosing *Escherichia coli* DH5α, *S. aureus* (SA), and *Salmonella enterica* Typhimurium (STM), with no significant differences identified (Fig 9A). When compared to other macrophage cell lines, J774A.1 cells showed significantly increased ability to phagocytose SA and RAW264.7 cells showed a significantly decreased ability to phagocytose STM. J774A.1 cells were also significantly better than RAW264.7 cells at phagocytosing DH5α (Fig 9B). The ability of macrophage models to phagocytose negatively charged carboxylated beads, mimicking apoptotic cells, was also assessed. Expectedly, *Msr1–/–* BMDMs displayed decreased ability to phagocytose the beads compared with WT BMDMs. Furthermore, each of the macrophage cell lines displayed significantly reduced phagocytic capacity, with iBMDMs and BMA3.1A7 cells phagocytosing the least (Fig 9C).

## The ability of different macrophage models to respond to pro- or anti-inflammatory stimuli varies

As differences in the expression of various proteins related to inflammation were highlighted in the proteomic comparison of

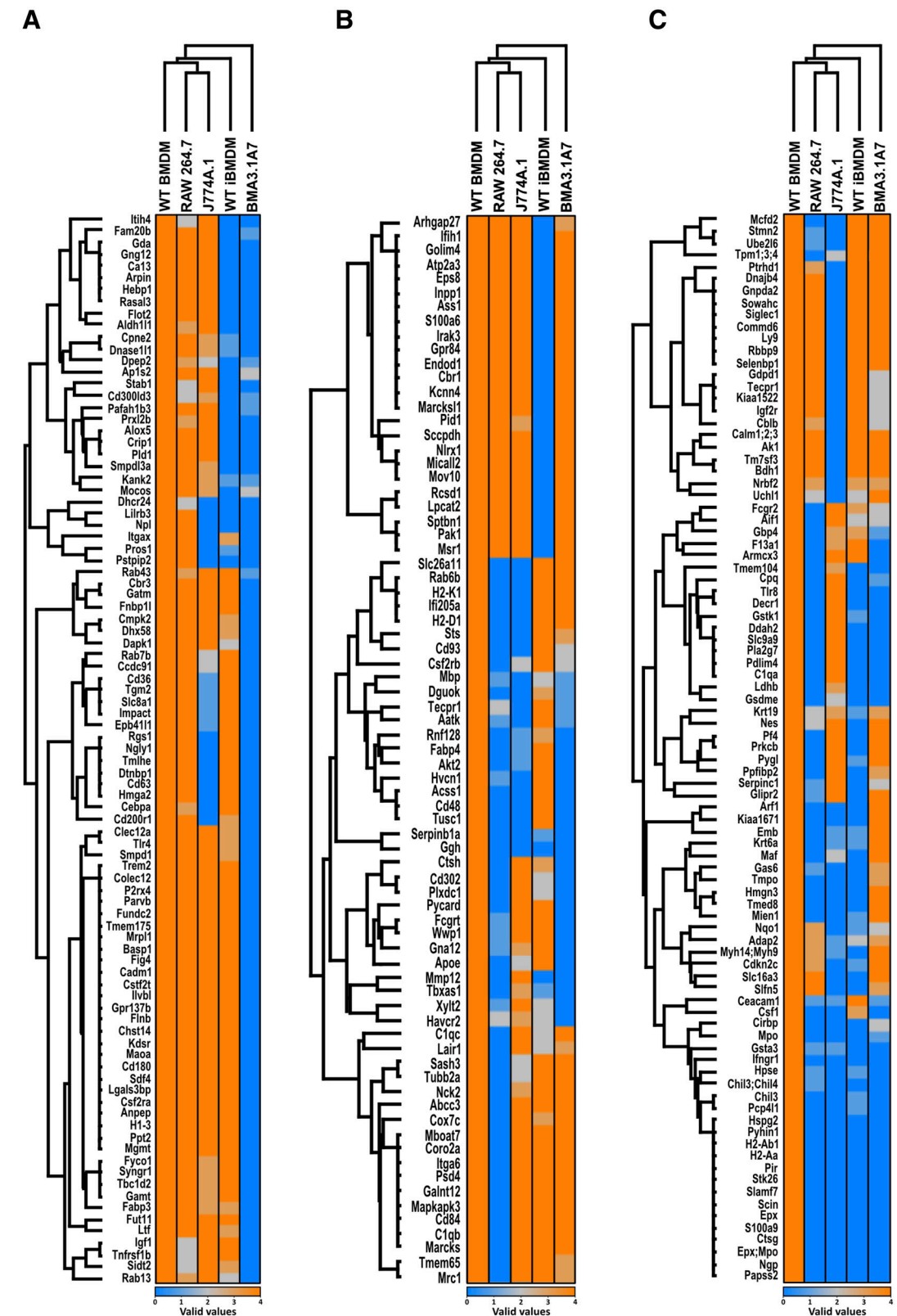

**Figure 7. Cell lines lack key proteins that are present in WT BMDMs.**
Presence/absence was determined using the number of valid values identified by mass spectrometry for each protein. Presence was defined as three or more valid values (orange), and absence was defined as zero valid values (blue). Only proteins displaying four valid values in WT BMDMs were included. **(A, B, C)** Hierarchical clustering was performed in Perseus with the resulting heatmap separated into three main clusters (A, B, C). N = 4.

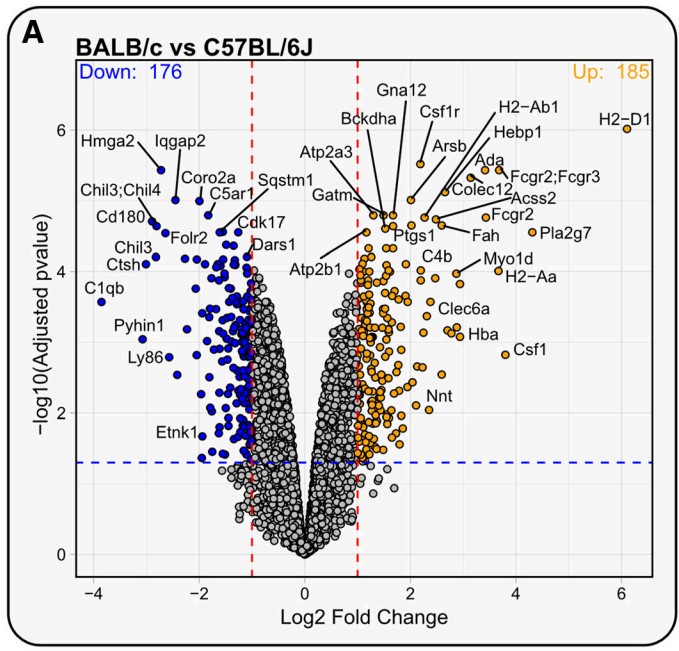

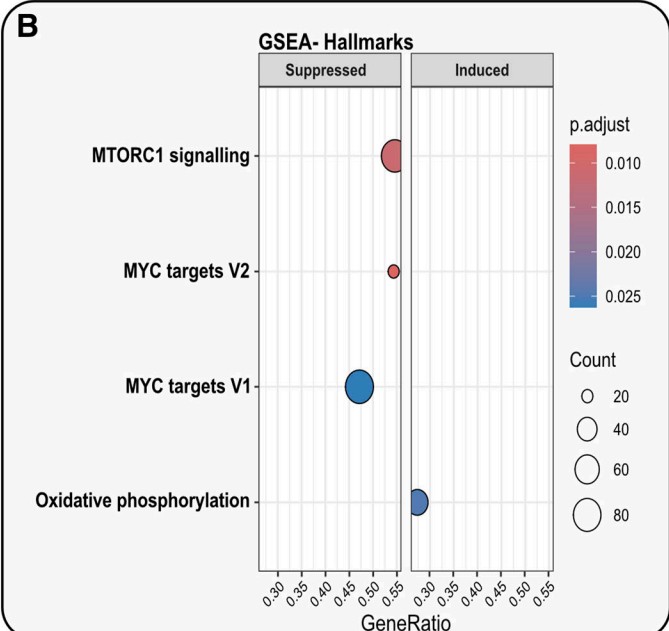

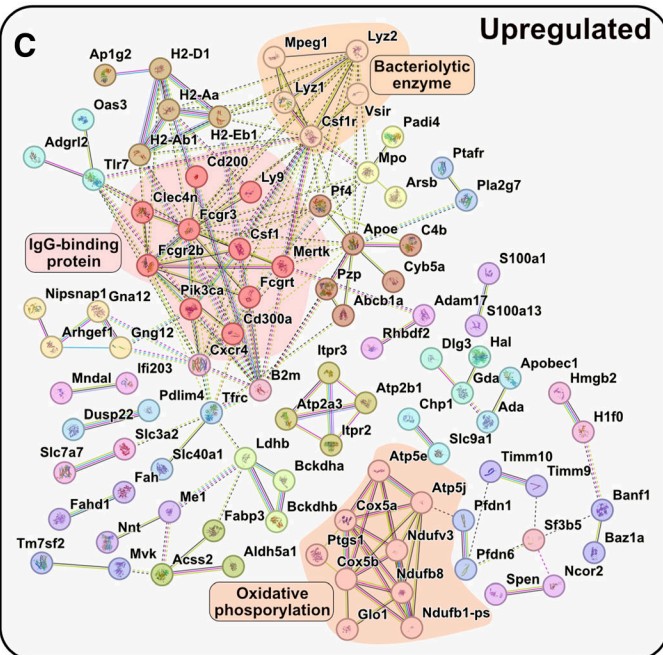

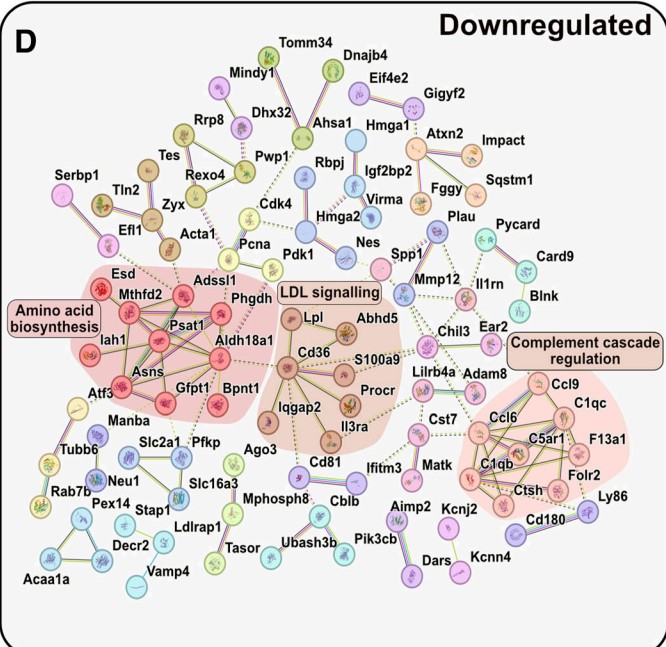

**Figure 8. Comparison of the proteomes of BALB/c and C57BL/6J BMDMs.**
**(A)** Volcano plot displaying proteomic data of BALB/c BMDMs versus C57BL/6J BMDMs (log$_2$ fold change > 1 or < −1; > 1.3 −log$_{10}$ [adjusted $P$-value]). Top 20 up- or down-regulated proteins are annotated. **(B)** GSEA of BALB/c BMDMs versus C57BL/6J BMDMs showing the suppressed or induced hallmark gene sets defined by MSigDB. **(C, D)** STRING analysis of the significantly up- and down-regulated proteins in BALB/c BMDMs versus C57BL/6J BMDMs; clusters were identified using the Markov Cluster Algorithm method.

macrophage cell lines with the gold standard C57BL/6J BMDM model, we sought to assess the response of macrophage models to different pro- or anti-inflammatory stimuli. M(IFN-γ) and M(LPS) macrophages increase the production of IL-1β, IL-6, and iNOS in response to activation, whereas M(IL-4) macrophages produce ARG1 (Murray et al, 2014).

IFN-γ stimulation of RAW264.7 cells induced significantly higher expression of *iNos* than all other macrophage models (Fig 10B).

However, no significant changes in *Il-1β* or *Il-6* expression were seen in any model (Fig 10A and C).

Post-LPS stimulation, all macrophage models besides J774A.1 displayed significantly lower expression of *Il-1β* when compared to C57BL/6J BMDMs (Fig 10A). BALB/c BMDMs exhibited a significant increase in *iNos* production, whereas iBMDMs, BMA3.1A7, and J774A.1 cells expressed significantly less than primary macrophage models and RAW264.7 cells (Fig 10B). There was also a significantly lower *Il-6*

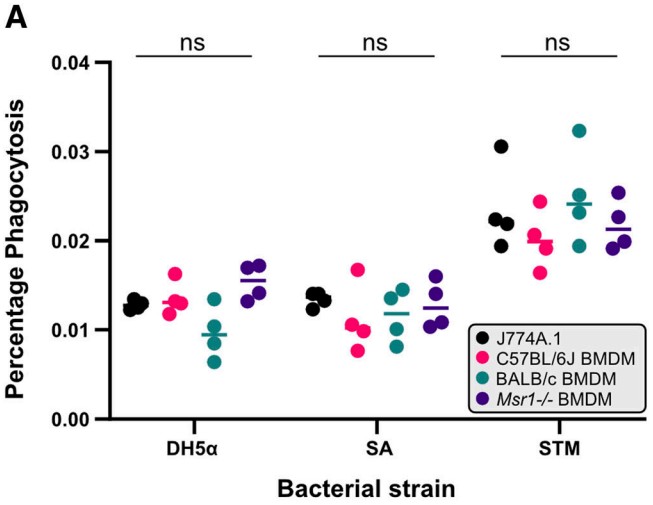

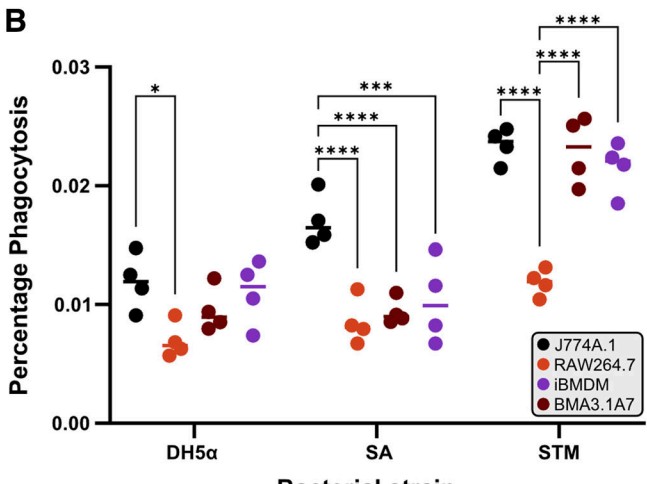

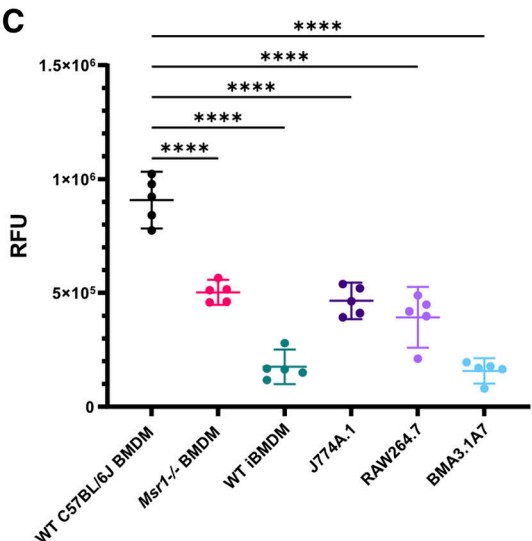

**Figure 9. Comparison of the phagocytic ability of macrophage cell lines compared with primary macrophages.**
**(A, B)** Percentage of phagocytosis of *Escherichia coli* DH5α (DH5α), *Salmonella enterica subsp. enterica* serovar Typhimurium (STM), and *Staphylococcus aureus* (SA) by cell lines (J774A.1, RAW264.7, iBMDM, BMA3.1A7) and primary macrophages isolated from C57BL/6J WT, *Msr1−/−*, and BALB/c mouse bone marrow. The assay measures CFUs of phagocytosed bacteria at 1-h post-infection of the respective cell lines or primary macrophages and is normalised to the CFUs in the pre-inoculum to calculate the percentage of phagocytosis. *$P < 0.05$; ***$P < 0.001$; ****$P < 0.0001$; ns, not significant; by ordinary two-way ANOVA. N = 4. **(C)** Phagocytosis of Alexa Fluor 488–labelled carboxylated beads by BMDMs from WT C57BL/6J and *Msr1−/−* BMDMs, and cell lines (J774A.1, RAW264.7 and BMA3.1A7, and iBMDM); the assay measures relative fluorescent units; trypan blue was used to quench the fluorescence from extracellular beads. ****$P < 0.0001$ by ordinary one-way ANOVA. N = 5.

response in BALB/c BMDMs, BMA3.1A7, J774A.1, and iBMDM cells. No significant difference was seen in IL-6 expression between RAW264.7 cells and C57BL/6J BMDMs (Fig 10C).

Compared with C57BL/6J BMDMs, IL-4 stimulation induced significantly lower expression of *Arg1* in BALB/c BMDMs, *Msr1−/−* BMDMs, iBMDM, BMA3.1A7, and RAW264.7 cells. Significant induction was seen in J774A.1 cells (Fig 10D).

## Discussion

The information provided here highlights the importance of cell line characterisation in comparison with the gold standard primary C57BL/6 BMDM model. Such characterisations enable selection of suitable models and insight into why differences in the response of cell lines to stimuli may differ.

Induction of common functions and pathways across all cell lines was highlighted, such as *Myc* activity, G2-M checkpoint, E2F gene transcription, and ribosome activity. This is indicative of the cellular changes needed to support continuous proliferation. MYC is a key transcription factor known to contribute to the development of several cancers. Specifically, *Myc* V1 targets include genes involved in cell cycle regulation, DNA replication, and protein synthesis, whereas V2 targets include genes involved in apoptosis and immune response. V1 targets are up-regulated by *Myc*, whereas V2 targets are down-regulated (Dang, 2012; Chen et al, 2018a). E2F

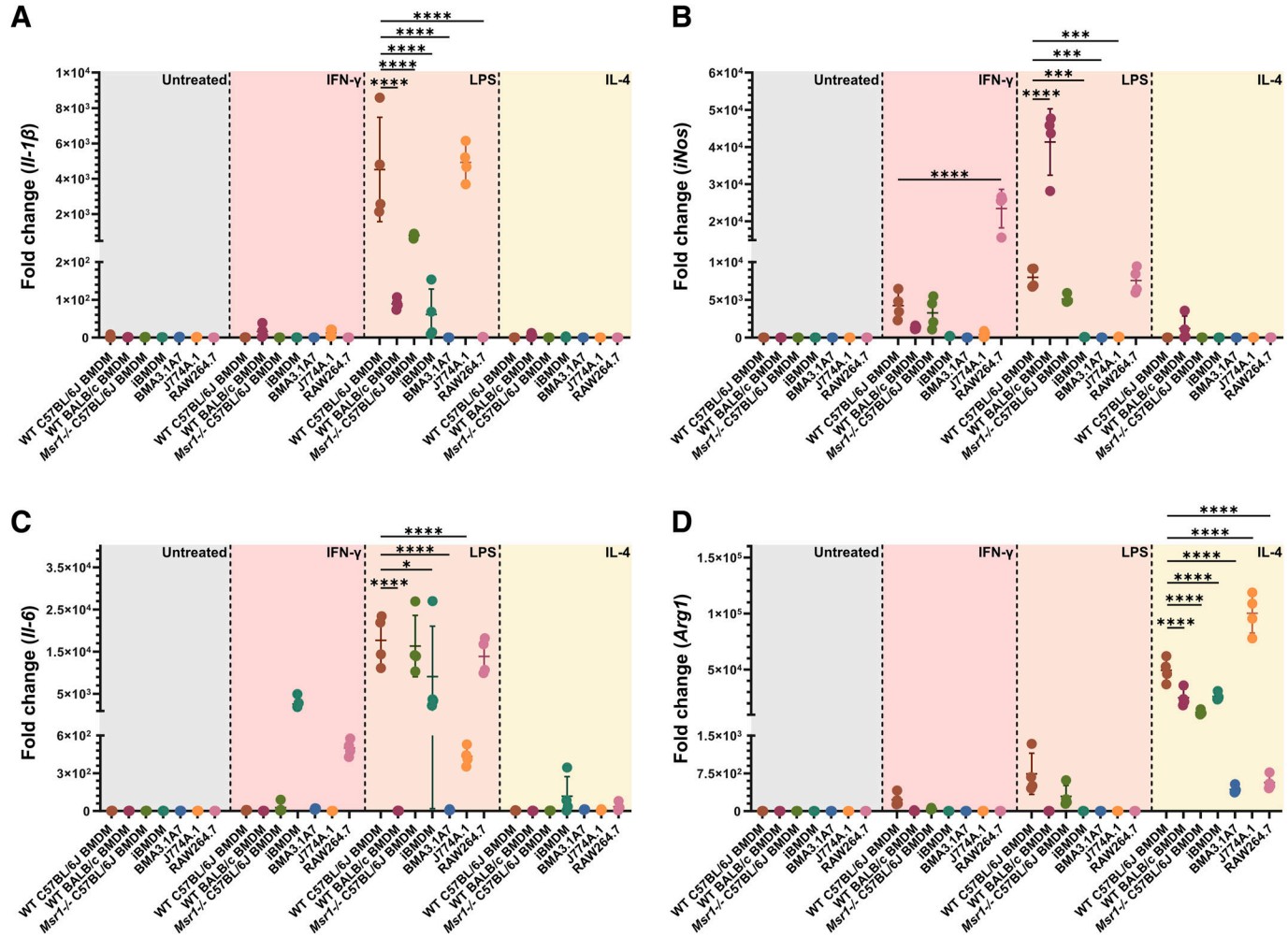

**Figure 10. Response to pro-inflammatory (IFN-γ and LPS) or anti-inflammatory (IL-4) stimuli differs between macrophage models.**
**(A, B, C, D)** Fold change in expression, measured by RT–qPCR, of (A) *Il-1β*, (B) *iNos*, (C) *Il-6*, and (D) *Arg1* after stimulation with IFN-γ (20 ng/ml, 16 h), LPS (100 ng/ml, 16 h), or IL-4 (20 ng/ml, 48 h). *P < 0.05; ***P < 0.001; ****P < 0.0001 by ordinary one-way ANOVA. N = 4.

targets and G2/M checkpoint also link to the immortalisation process and aberrant cell cycle progression (Oshi et al, 2020a, 2020b).

Complement and coagulation cascade–related proteins were also seen to be down-regulated in each cell line. These cascades form part of the first line of innate defence against invading pathogens in circulation, with both systems relying on complex enzymatic signalling cascades for efficient activity. The same changes were seen in a proteome comparison of the hepatoma cell line Hepa1-6 with primary hepatocytes (Pan et al, 2009). Therefore, it is likely that loss of tissue context means that it is no longer necessary for the cell lines to produce these proteins. However, besides these common differences, each cell line displayed varied changes relating to normal macrophage function and innate immunity.

The J2 immortalisation process induced changes mainly related to macrophage polarisation. The pro-inflammatory JAK/STAT pathway was suppressed in iBMDMs; this pathway is linked with increased tumour cell proliferation and dampened antitumour immune response. In the tumour microenvironment, the IL-6/JAK/STAT3 axis is stimulated by release of IL-6 from cells such as TAMs (Johnson et al, 2018). Furthermore, proteins relating to oxidative phosphorylation were significantly induced. M(IL-4) macrophages rely on oxidative phosphorylation for energy metabolism, whereas M(LPS) and M(IFN) macrophages rely mainly on glycolysis (Viola et al, 2019). The reduction in complement activity seen in these cells, specifically activity of C5a and C3a, may also be partially responsible for the suppression of the IL-6/JAK/STAT3 cascade, as C5a and C3a are known to stimulate IL-6 release in Kupffer cells (Oikonomopoulou et al, 2012). Further decreased ability to respond to and clear pathogens was indicated by the down-regulation of Fc-γ–mediated phagocytosis and Lyz1/2. Fcγ receptors on macrophages mediate the uptake and killing of foreign materials such as antibody-coated bacteria, viruses, and parasites, as well as host cells that express viral or tumour antigens (Fitzer-Attas et al, 2000), whereas Lyz1/2 are important enzymes capable of hydrolysing bacterial cell wall peptidoglycans (Ragland & Criss, 2017).

J774A.1 cells were determined to be the most similar to WT C57BL/6J BMDMs, despite being a BALB/c model. However, the cell line harboured some important changes. Knockdown of the most down-regulated protein in J774A.1 cells, GPNMB, and in RAW264.7 cells resulted in increased levels of pro-inflammatory cytokines, whereas overexpression dampened the pro-inflammatory response after IFN-γ/LPS stimulation (Saade et al, 2021). MRC1, also down-regulated, plays multiple roles in macrophage activity, including the recognition and endocytic clearance of both endogenous and pathogen-related ligands, antigen processing and presentation, and TLR-2 signalling (van der Zande et al, 2021). Furthermore, ablation of the receptor increases levels of pro-inflammatory proteins. Similarly, CD9 has been shown to negatively regulate LPS-induced macrophage activation (Brosseau et al, 2018; Orecchioni et al, 2019). The down-regulation of these proteins suggests a shift towards pro-inflammatory activity. Several S100 proteins were also down-regulated. The S100 proteins hold a range of functions but have importantly been linked with the regulation of macrophage inflammation. S100a6 interacts with receptor for advanced glycation end products to mediate cell survival and apoptosis and activates pro-inflammatory JNK. S100A10 has been shown to mediate the migration of macrophages towards tumour sites and contribute to MAPK and NF-κB induction of inflammatory cytokines. S100A11 is an alarmin, an endogenous chemotactic and immune-activating protein secreted in response to *Toxoplasma gondii*. S100A13 also further contributes to cytokine and antimicrobial agent release (Xia et al, 2018; Singh & Ali, 2022).

Changes seen in RAW264.7 cells were more varied. MPEG1 was down-regulated; this is normally induced in response to pro-inflammatory stimuli, with its loss linked to increased susceptibility to bacterial infection (Bayly-Jones et al, 2020). In addition, suggesting decreased pro-inflammatory ability, IGF2R was up-regulated. Activation of IGF2R reprograms cellular metabolism towards oxidative phosphorylation and, thus, an anti-inflammatory phenotype (Wang et al, 2020). However, also implicated in metabolic reprogramming, SLC2A1 (GLUT1) was up-regulated. This is the main rate-limiting glucose transporter found in pro-inflammatory macrophages. Such macrophages favour glucose as an energy substrate; therefore, RAW264.7 cells may more readily acquire a pro-inflammatory metabolic phenotype (Freemerman et al, 2014). Further supporting increased pro-inflammatory activity in RAW264.7 cells, cellular nucleic acid–binding protein is known to regulate a specific gene signature in macrophages by controlling the activity of c-Rel and its translocation to the nucleus. This activity specifically controls the induction of IL-12β expression and therefore impacts the T helper type 1 and IFN-γ–mediated immune response against a range of pathogens (Chen et al, 2018b).

BMA3.1A7 cells harboured the greatest number of significant changes in proteins linking strongly to normal macrophage function, with many down-regulated or not detected. Firstly, MPEG1 was down-regulated 256-fold, potentially increasing susceptibility to bacterial infection by decreasing the pro-inflammatory response (Bayly-Jones et al, 2020). CD84 has been implicated in the LPS response by rapidly increasing phosphorylation of ERK1/2, p38, and JNK-1/2; and by activating NF-κB (Sintes et al, 2010). Down-regulation may therefore hinder the ability to fully respond to LPS. TLR-13 recognises a conserved motif present in 23S ribosomal

RNA from both Gram-positive and Gram-negative bacteria (Li & Chen, 2012), as well as vesicular stomatitis virus (Shi et al, 2011). Group B streptococcus was used to demonstrate that upon ligand binding, TLR-13 mediates TNF-α and nitric oxide induction (Signorino et al, 2014). Lyz1 and Lyz2 were again seen to be significantly down-regulated, limiting protection against invading pathogens (Ragland & Criss, 2017). CD206, commonly referred to as an anti-inflammatory macrophage marker that recognises various PAMPs, was also down-regulated (Abdelaziz et al, 2020). Galectin-3 inhibition can reduce inflammatory response and the expression of markers such as TNF-α and IL-1β (Lu et al, 2020), whereas galectin-9 can stimulate monocyte differentiation towards an anti-inflammatory phenotype (Enninga et al, 2016). Further to significantly down-regulated proteins, several key proteins were not identified at all in BMA3.1A7 cells. STAB1 acts as an endocytic receptor for self-ligands such as acLDL and as a phagocytic receptor for apoptotic bodies (Kzhyshkowska, 2010). Similarly, CD36 has a high affinity for specific phospholipid moieties found on oxLDL and apoptotic cells and also binds glycated proteins, amyloid β, LTA from bacteria, and glycans from fungi. Recognition of these DAMPs or PAMPs results in a pro-inflammatory response mediated by NF-κB (Chen et al, 2022). COLEC12 may be responsible for the clearance of amyloid β (Kelley et al, 2014). TGM2 is known to promote efferocytosis, with ablation of the enzyme reducing levels of efferocytic receptors CD14 and MSR1, and promoting pro-inflammatory signalling (Eligini et al, 2016). TREM2 is a phagocytic receptor for bacteria and amyloid β (Li & Zhang, 2018). Finally, TLR-4 is the major TLR for LPS and other exogenous ligands, implicating the receptor in host defence against Gram-positive and Gram-negative bacteria, viruses, and fungi. TLR-4 also recognises endogenous ligands such as heat-shock proteins, hyaluronan, and amyloid β. The loss of these proteins in BMA3.1A7 cells, among the many others outlined in Fig 7, indicates that this cell line has an impaired ability to recognise exogenous PAMPs and endogenous DAMPs. Therefore, this cell line may not be suitable for experiments focused on polarisation, phagocytosis, efferocytosis, or infection models such as *S. aureus* infection.

Investigation of receptor expression highlighted that J774A.1 cells hold the most similar receptor profile to WT BMDMs. Mer tyrosine kinase (MERTK) was only identified in J774A.1 cells; this receptor plays an important role in the regulation of cytokine secretion and apoptotic cell clearance. Furthermore, mice deficient in MERTK are hypersensitive to LPS-induced endotoxic shock, indicating that the receptor has a role in dampening the immune response in macrophages (Anwar et al, 2009). J774A.1 cells therefore may be a more suitable model for exploring the effect of LPS treatment on macrophage function. The expression of TLR-7 and TLR-8 was also most similar to WT BMDMs in J774A.1 cells; both receptors recognise single-stranded RNA and induce an inflammatory response (Eng et al, 2018). This means that these cells may give a more representative response to single-stranded RNA viruses than other models. FCGR2 expression is also normal in J774A.1 cells compared with RAW264.7 cells, indicating that J774A.1 cells may be a better model for investigating the role of macrophages in the phagocytosis and killing of opsonised pathogens.

One of the more puzzling findings was the lack of identification of IFNGR1 in all cell lines. IFN-γ signals by first binding to its dimeric

receptor consisting of IFNGR1 and IFNGR2. This activates the JAK-STAT signalling pathway and IFN-stimulated gene (ISG) production. ISGs encode a wide range of products that hold vital immune effector functions such as antigen presentation, chemokine activity, and phagocytosis. Therefore, IFN-γ is one of the most important endogenous inflammatory and immune mediators, vital for host defence against intracellular pathogens (Hu & Ivashkiv, 2009; Ivashkiv, 2018). The lack of identification of this vital receptor across all cell lines was unexpected as each cell line has previously been reported to respond to IFN-γ stimulation (Raso et al, 2002; Elizabeth et al, 2016; Gao et al, 2016; Bilkei-Gorzo et al, 2022). Indeed, flow cytometry analysis confirmed the presence of the IFNGR complex on the surface of all macrophage models, demonstrating the importance of validation of mass spectrometry results, especially when proteins are only quantified with few peptides (three peptides for IFNGR1) and the combined protein intensities are close to the detection level.

The caveat to the main proteomic comparisons performed here is that all cell line comparisons were made to C57BL/6J BMDMs. Both J774A.1 and RAW264.7 cells were obtained from BALB/c mice, with J774A.1 cells generated from ascites of reticulum cell sarcoma (Ralph et al, 1976) and RAW264.7 cells produced by immortalisation with Abelson murine leukaemia virus (Raschke et al, 1978). Therefore, differences at the proteome level may be influenced by strain-specific genetic differences. However, macrophage investigations throughout the literature use either BMDMs from C57BL/6 mice or J774A.1 and RAW264.7 cells. Therefore, these comparisons remain valuable. Further differences may also be a result of the method used to generate the cell lines. The J774A.1 cell line displayed the least amount of change relevant to macrophage function and is the only line to not be generated using a virus. This potentially indicates that tumour-derived macrophage cell lines may be more similar to primary cells than cell lines generated using oncogenic viruses.

To further investigate the differences observed between macrophage cell lines of different origins and the gold standard primary C57BL/6J macrophage model, we conducted a separate mass spectrometry–based proteomic experiment to compare BMDMs from BALB/c and C57BL/6J mice. This analysis identified fewer significant changes compared with comparisons between cell lines and C57BL/6J BMDMs, yet still highlighted some potentially important differences. Both CSF1 and CSF1R were significantly up-regulated in BALB/c BMDMs, potentially influencing the differentiation and proliferation of these macrophages (Sehgal et al, 2021). Multiple proteins with pattern recognition capabilities were up-regulated. COLEC12 is a class A scavenger receptor able to mount an immune response to bacteria and yeast (Taban et al, 2022). This receptor was up-regulated in all cell lines compared with C57BL/6J BMDMs except for in BMA3.1A7 cells, in which it was not identified. CLEC6A, part of the C-type lectin receptor family, is a PRR for a range of pathogens including fungi and *Mycobacterium tuberculosis* (Graham & Brown, 2009). The chemokine receptor CXCR4 binds to LPS and induces inflammatory cytokine expression (Tian et al, 2019). MERTK, previously only identified in J774A.1 cells and linked to negative regulation of LPS signalling, was also up-regulated in BALB/c BMDMs. Hydrolytic enzymes LYZ1 and LYZ2 were up-regulated; this is in contrast to all cell lines when compared to C57BL/6J BMDMs, potentially indicating that

immortalisation processes drive down the expression of these enzymes. Interestingly, a collection of proteins all linked to the TLR-4 mediated LPS response was down-regulated. CD180 and its accessory molecule LY86, better known as MD-1, are known to interact with TLR-4 after challenge with LPS or *M. tuberculosis* (Schultz & Blumenthal, 2017). Similarly, scavenger receptor CD36 is able to bind to a wide range of endogenous and exogenous polyanionic ligands and form a complex with TLR-4 and TLR-6 to induce pro-inflammatory signalling (Stewart et al, 2010). Overall, taken along with the up-regulation of proteins related to oxidative phosphorylation, the changes in protein expression relating to pro-inflammatory signalling may in part explain why BALB/c mice more readily mount a Th2 response compared with C57BL/6J mice (Mills et al, 2000). RNA-sequencing analysis of BMDMs from five different mouse strains has previously highlighted that BALB/c BMDMs exhibit the lowest level of divergent gene expression compared with C57BL/6 BMDMs than the other common strains (Link et al, 2018). The fewer differences observed between the two mouse genotypes indicate that the changes seen between cell lines and C57BL/6J BMDMs may be influenced more by the immortalisation process used to generate the models rather than the genotype of the cell.

Functional validation of the phagocytic ability of different macrophage models revealed that J774A.1 cells may be the most suitable cell line model for investigations surrounding the phagocytosis of bacteria or apoptotic cells. This is in agreement with the proteomic analysis, which highlighted no changes in proteins or gene sets related to phagocytosis. In contrast, RAW264.7 cells, in which GSEA highlighted suppression of terms such as "phagosome" and "bacterial invasion" when compared to primary BMDMs, performed the worst. The decreased ability of *Msr1−/−* BMDMs, iBMDMs, and BMA3.1A7 cells to phagocytose negatively charged carboxylated beads may be related to the loss or decreased expression of MSR1. MSR1 can act as a phagocytic receptor but only binds ligands that hold multiple negative charges; therefore, the loss of the receptor may impact the uptake of negatively charged ligands such as apoptotic cells (Guo et al, 2019). The TLR2/4 receptors are well documented to be one of the prime microbial ligands mediating phagocytosis of SA and DH5α, and down-regulation of these receptors is directly linked to reduced phagocytic activity, fitting our proteomic data (Baranova et al, 2008; Fu & Harrison, 2021). Interestingly, the overall phagocytosis of STM is significantly higher than DH5α and/or SA; this can be attributed to the fact that STM can mediate its uptake via *Salmonella* pathogenicity island–encoded type 3 secretion system (T3SS) effectors (Drecktrah et al, 2006).

To assess the ability of different macrophage models to polarise towards M(IFN-γ), M(LPS), and M(IL-4) phenotypes, RT–qPCR analysis was carried out focused on different pro-inflammatory (*Il-1β*, *iNos*, and *Il-6*) or anti-inflammatory (*Arg1*) markers. This revealed a great degree of variation in response. LPS-induced *Il-1β* expression in BALB/c BMDMs was significantly lower than C57BL/6J BMDMs and was not seen in RAW264.7 cells; however, J774A.1 cells produced *Il-1β* at levels similar to C57BL/6J BMDMs. As these cell lines are both of BALB/c origin, this indicates that genetic background alone may not determine this M(LPS) response and that J774A.1 cells may be the most suitable cell line for investigating *Il-1β*–driven pro-inflammatory responses. However, the opposite effect was seen for LPS-induced *iNos* expression, with BALB/c BMDMs displaying significantly higher induction

than C57BL/6J BMDMs, RAW264.7 cells showing no significant difference, and J774A.1 cells being unresponsive. Similar results were seen post–IFN-γ stimulation. Again, *Il-6* induction post-LPS stimulation is significantly lower in J774A.1 cells compared with other models. Contrary to this, post–anti-inflammatory IL-4 stimulation, J774A.1 cells responded with the greatest induction of *Arg1*, with RAW264.7 cells responding with a significantly lower induction. BMA3.1A7 cells mounted a much lower response to any stimulus compared with the other cell lines across the inflammatory markers tested. The lack of response by BMA3.1A7 cells corroborates the proteomic analysis, which indicated that BMA3.1A7 cells are the least similar to the primary BMDM model; however, the reduced response of J774A.1 cells in most conditions is in contradiction to the proteomic results discussed. Further to this, although the expression of IFNGR1 and IFNGR2 was lowest on RAW264.7 cells, this cell line responded the best to IFN-γ stimulation, indicating receptor expression alone is not enough to assess ligand response. Altogether, the induction of inflammatory markers by different models is complex and highlights the need for testing in different models.

It appears that the process of J2 immortalisation drove the unexpected loss of MSR1, as confirmed by mass spectrometry, flow cytometry, and RT–qPCR. This has not been noted in any previous literature. The molecular reasoning behind the loss of MSR1 is unclear; however, as RT–qPCR analysis indicated diminished gene transcription, it is likely to be a result of genetic or epigenetic interference. One potential mechanism of this is gene disruption because of transgene insertion. As the J2 virus is a gamma-retrovirus, it may have an increased propensity for insertion near promoter regions (Baum et al, 2004). Alternatively, the J2 virus may have induced CpG methylation near the transcription start site or promoter regions of *Msr1*, which could silence the gene. The same mechanism may be responsible for the silencing of other genes discussed here.

It is hoped that the datasets presented here (Tables S1 and S2) will be of valuable use to the macrophage biology community. The differential expression of various proteins may explain experimental differences obtained when comparing primary models to cell line models. Furthermore, the proteome profiles generated can guide the selection of optimal models for use in experiments covering areas such as phagocytosis, polarisation, and infection.

# Materials and Methods

### Cell culture

J774A.1 (TIB-67), RAW264.7 (TIB-71), AMJ2-C11 (CRL-2456), and L929 (CCL-1) cell lines were purchased from the American Type Culture Collection (ATCC, Manassas, VA, USA). The BMA3.1A7 (accession: CVCL_IW58) was kindly provided by Kenneth Rock (Dana Farber Centre, Boston, US). Cells were maintained in medium at 37°C in a humidified 5% $CO_2$ atmosphere. The ATCC routinely performs cell line authentication, using short tandem repeat profiling as a procedure. Cell experimentation was always performed within a period not exceeding two months after resuscitation, and cells were regularly tested for mycoplasma.

### Mice

WT C57BL/6J and BALB/c mice were obtained from Charles River. $Msr1^{-/-}$ mice were kindly provided by Siamon Gordon. Newcastle University Ethical Committee approved animal work, and manipulation was performed under UK Home Office project licence.

### Generation of murine BMDMs

Bone marrow cells were collected from femurs and tibiae of 8- to 12-wk-old WT and $Msr1^{-/-}$ C57BL/6J mice. Collected cells were treated with red blood cell lysis buffer (155 mM $NH_4Cl$, 12 mM $NaHCO_3$, 0.1 mM EDTA) and plated on untreated 10-cm cell culture dishes (BD Biosciences) in IMDM (Gibco) containing 10% heat-inactivated FBS, 100 U/ml penicillin/streptomycin (Gibco), and 15% L929 conditioned supplement. After 24 h, the cells in the supernatant were transferred to untreated 10-cm Petri dishes (BD Biosciences) for 7 d for the differentiation into BMDMs (Heap et al, 2021).

### Immortalisation of murine BMDMs

The AMJ2-C11 cell line was used as the source of the J2 virus needed for the immortalisation of the target cell lines. AMJ2-C11 cells were cultured in complete DMEM (AMJ2-C11) until 80% confluent. Then, 75% of these cells were seeded using bone marrow growth medium and incubated for 24 h. The J2 virus–containing conditioned media were then decanted and clarified by centrifugation at 200*g* for 5 min, followed by filtration through a 0.45-*μ*m filter. Aliquots were either used immediately to avoid loss of viral titre or stored at −20°C.

Murine BMDMs were generated as previously described and transduced twice with the J2 virus as follows. Five days after harvest, bone marrow growth media were replaced with 50% J2 conditioned media and 50% bone marrow growth media for 24 h, keeping a control dish of cells for comparison. After 24 h, J2 conditioned media were removed, and the cells were allowed to recover for a further 24 h in fresh media. The first transduction step was then repeated for a final 24 h. Cells were routinely monitored for growth or death compared with the control and passaged accordingly. 1 wk after transduction, the concentration of L929 conditioned medium in the bone marrow growth media was lowered from 20% to 10% with the aim of reducing this concentration to 0% over the next 2–3 mo. This process is likely being carried out to wean the cells off the dependency on L929 conditioned medium. If the iBMDMs ceased to grow after a reduction in L929 concentration, the concentration was returned to its previous level and the cells were allowed to recover before attempting to lower the concentration again.

### RT–qPCR

RNA extraction was performed using QIAGEN RNeasy Mini columns and QIAshredder for homogenisation kit (QIAGEN) following the manufacturer's instructions. RNA was quantified using a NanoDrop spectrophotometer. Reverse transcription was performed with 1 *μ*g of template RNA using the QIAGEN QuantiTect reverse transcription

kit (QIAGEN) in Bio-Rad T100 Thermal Cycler (Bio-Rad Laboratories) to obtain complementary DNA (cDNA). cDNA was diluted to 5 ng/$\mu$l in RNase-free water, and 1 $\mu$l was used for RT–qPCR with QIAGEN QuantiNova SYBR Green Master Mix (QIAGEN) in StepOnePlus RT–qPCR System (Applied Biosystems) according to the manufacturer's instructions. Each sample was run in triplicate and the 2-ΔΔCt method used to calculate the relative expression of genes (Table 2). Expression values of *Gapdh* and *Tbp* genes were used for normalisation.

## Proteomic sample preparation

The whole-cell protein lysate was prepared by sonication in SDS lysis buffer (5% SDS, 50 mM TEAB, pH 8.5, in HPLC water supplemented with cOmplete, Protease Inhibitor Cocktail [Sigma-Aldrich]). Protein concentration was quantified using Pierce BCA Protein Assay (Thermo Fisher Scientific) according to the manufacturer's instructions and measured at 562 nm using the SpectraMax iD3 microplate reader (Molecular Devices). 25 $\mu$g of each sample was reduced by addition of Tris(2-carboxyethyl) phosphine (TCEP; Pierce) to a final concentration of 10 mM for 30 min at 37°C, and subsequently alkylated with iodoacetamide (IAA) to a final concentration of 10 mM for 30 min at room temperature in the dark. Samples were acidified by the addition of 2.5 $\mu$l of 12% phosphoric acid and diluted with 165 $\mu$l of S-trap binding buffer (90% MeOH, 100 mM TEAB, pH 7.1). The acidified samples were then loaded onto the S-trap spin column (ProtiFi) and centrifuged at 4,000*g* for 1 min. Each S-trap mini-spin column was washed with 150 $\mu$l S-trap binding buffer and centrifuged at 4,000*g* for 1 min; this was repeated five times with rotation of the column between each wash. For protein digestion, TEAB (50 mM, pH 8.0) containing sequencing-grade trypsin (1:10 ratio of trypsin:protein) was added to each sample, followed by incubation for 2 h at 47°C using an unmoving thermomixer (Eppendorf). Peptides were eluted with 40 $\mu$l TEAB (50 mM, pH 8.0) and centrifugation at

1,000*g* for 1 min. Elution steps were repeated using 40 $\mu$l formic acid (0.2%) and, finally, 35 $\mu$l formic acid (0.2%), and acetonitrile (50%). Eluates were combined and dried down using a speed-vac before storage at –80°C.

## High-performance liquid chromatography

Dried peptide samples were resuspended in MS sample buffer (acetonitrile [2% vol/vol], trifluoroacetic acid [0.1% vol/vol] in HPLC-grade water) to a concentration of 500 ng/$\mu$l and placed into glass autosampler vials. Peptide samples were injected on a Dionex UltiMate 3000 RSLC (Thermo Fisher Scientific), connected to an Orbitrap QE HF mass spectrometer (Thermo Fisher Scientific), using a PepMap 100 C18 LC trap column (300 $\mu$m ID × 5 mm, 5 $\mu$m, 100 Å). This was followed by separation on an EASY-Spray column (50 cm × 75 $\mu$m ID, PepMap C18, 2 $\mu$m, 100 Å) (Thermo Fisher Scientific). Peptides were separated using a linear gradient of 3–35% Buffer B (acetonitrile [80% vol/vol], formic acid [0.1% vol/vol] in HPLC-grade water; Buffer A: formic acid [0.1% vol/vol] in HPLC-grade water) over 120 min for whole-cell samples, followed by a step from 35–90% Buffer B in 30 s at 250 nl/min, and held at 90% for 4 min. The gradient was then decreased to 3% Buffer B in 0.5 min at 250 nl/min for 20 min. The column temperature was controlled at 45°C. For the mouse strain comparison, LC was performed on 500 ng of peptides using an Evosep One system with a 15-cm Aurora Elite C18 column with integrated captive spray emitter (IonOpticks), at 50°C. Buffer A was 0.1% formic acid in HPLC water, and buffer B was 0.1% formic acid in acetonitrile. The Whisper-Zoom 20 SPD protocol was used in this study (where the gradient is 0–35% Buffer B, 200 nl/min, for 58 min, and 20 samples per day were permitted).

## Data-independent acquisition LC-MS/MS analysis

Whole-cell proteomes were analysed using DIA on an Orbitrap QE HF mass spectrometer. The QE HF mass spectrometer was operated in positive ion, DIA mode. Full MS scan spectra were acquired in the m/z range of 400–1,300 with an automatic gain control target of 1 × 10$^6$ and a maximum injection time of 60 ms, at a resolution of 120,000. Targeted MS2 scan spectra were acquired in the m/z range of 400–1,000 using 16 m/z quadrupole isolation windows, AGC target of 1 × 10$^6$, at a resolution of 30,000, maximum injection time of 55 ms, and loop count of 45. Higher energy collision-induced dissociation fragmentation was performed in one-step collision energy of 27%. Electrospray voltage was static, and capillary temperature was 275°C, with expected LC peak width of 30 s. No sheath and auxiliary gas flow were used. For the mouse strain comparison, analysis was performed using a timsTOF HT (Bruker), operated in the DIA-PASEF mode. Mass and IM ranges were 300–1,200 *m/z* and 0.6–1.45 1/$K_0$, DIA-PASEF was performed as previously described (Hermosilla-Trespaderne et al, 2024) using variable width IM-*m/z* windows without overlap, designed using py_diAID (Skowronek et al, 2022). TIMS accumulation and ramp times were 66 ms, and total cycle time was ~1.2 s. Collision energy was applied in a linear fashion, where ion mobility = 0.6–1.6 1/$K_0$, and collision energy = 20–59 eV.

**Table 2. List of primers.**

| Target gene | Direction | Sequence (3′ À 5′) |
|---|---|---|
| *Msr1* | Forward | TGAACGAGAGGATGCTGACTG |
| | Reverse | TGTCATTGAACGTGCCGTCAAA |
| *Gapdh* | Forward | CATCACTGCCACCCAGAAGACTG |
| | Reverse | ATGCCAGTGAGCTTCCCGTTCAG |
| *Tbp* | Forward | CCCCTTGTACCCTTCACCAAT |
| | Reverse | GAAGCTGCGGTACAATTCCAG |
| *Il-6* | Forward | ACAAAGCCAGAGTCCTTCAGAGA |
| | Reverse | AGGAGAGCATTGGAAATTGGGGT |
| *iNos* | Forward | GACACAGTGTCACTGGTTTG |
| | Reverse | TTTGAAGAGAAACTTCCAGG |
| *Il-1β* | Forward | TGCCACCTTTTGACAGTGATGA |
| | Reverse | TGCCCTGGGGAAGGCATTAG |
| *Arg1* | Forward | CTCCAAGCAAAGTCCTTAGAG |
| | Reverse | AGGAGCTGTCATTAGGGACATC |

**Table 3. Flow cytometry antibodies.**

| Antibody-conjugate | Supplier | Catalogue no. | Host species |
|---|---|---|---|
| F4/80-Alexa Fluor 488 | Invitrogen | 53-4801-82 | Rat |
| CD11B-BUV737 | Invitrogen | 367-0112-82 | Rat |
| CD11C-APC | Invitrogen | 17-0114-82 | Armenian hamster |
| MSR1-PE | Invitrogen | 12-2046-82 | Rat |
| IFNGR1-PE | BioLegend | 113605 | Armenian hamster |
| IFNGR2-APC | Thermo Fisher Scientific | 12-1191-82 | Armenian hamster |

### Data-independent acquisition protein identification using DIA-NN

Raw files were searched in DIA-NN version 1.8 against a UniProt *Mus musculus* database (containing 25,387 entries, downloaded 15/03/21), as well as a common contaminants database (Frankenfield et al, 2022), at 1% false discovery rate with default settings. An in silico digest for spectral library generation was used for library-free search.

### Proteomic data analysis

Proteins were first filtered to remove contaminants and those identified by only one unique peptide. For whole-cell proteome analysis, Perseus V2.0.7.0 was used to generate heatmaps with hierarchical clustering and GO-term enrichment according to the protocol outlined by Tyanova and Cox (Tyanova & Cox, 2018). RStudio was used to $\log_2$-transform and median-normalise the data, and determine which proteins showed significantly differential expression (adjusted $P < 0.05$ and fold change $\geq 2$ or $\leq -2$) using the limma package with the Benjamini–Hochberg correction used for multiple comparison testing (Ritchie et al, 2015). GSEA was performed using the clusterProfiler package with a minimum and maximum gene set size of 20 and 600, respectively. The Benjamini–Hochberg correction was used for multiple comparisons with a $P$-value cut-off of 0.05. Both hallmark gene sets defined by the Molecular Signatures Database (MSigDB) and gene sets defined by the Kyoto Encyclopaedia of Genes and Genomes (KEGG) were used for GSEA (Wu et al, 2021). STRING network analysis was performed with a minimum required interaction score of 0.4 using the following interaction sources: text mining, experiments, databases, and co-expression. Disconnected nodes were hidden from the visual network but included in the inbuilt enrichment analysis.

### Bacterial uptake assay

The bacterial culture of *E. coli* DH5α (DH5α) and *S. enterica subsp. enterica* serovar Typhimurium (STM) was maintained in Luria–Bertani (LB) media, and *S. aureus* (SA) was maintained in tryptic soy broth/agar (TSB/TSA).

For each experimental replicate, $1 \times 10^5$ cells were seeded per well of 24-well tissue culture dishes. Before infection, the cells were washed three times with warm PBS, and a fresh, complete medium was added to each well. Overnight, bacterial culture of DH5α, STM,

and SA was subcultured to the mid-exponential phase (optical density (OD600 nm) of 0.6–0.7). After 2 h, when the OD600 reached 1 ($1 \times 10^9$ bacteria/ml), bacteria were washed in warm PBS twice at 2,500*g* for 15 min and resuspended in a warm cell culture medium. The bacterial pre-inoculum was added at an MOI of 1:10 and incubated for 30 min for uptake. After 30 min, the bacterium-containing medium was removed, cells were washed twice with warm PBS, and medium containing 50 μg/ml gentamicin was added. After 1 h, CFUs were determined by lysing cells in 0.1% Triton X-100 in PBS. After serial dilution, the lysates and pre-inoculum were plated onto agar culture plates, and CFU/ml was calculated.

$$\% \text{ Phagocytosis} = \text{CFU at 1 h/CFU of pre-inoculum}*100.$$

### Bead phagocytosis assay

The assay was performed similar to the previously published work (Méndez-Alejandre et al, 2023). Cells were seeded at a density of $10^5$ per well of a 96-well plate and left to adhere for 24 h. Alexa Fluor 488–labelled carboxylated beads in PBS were added for 10 min, then washed off. Trypan blue was used to quench the fluorescence of extracellular beads, and the relative fluorescent units were measured using a plate reader.

### Flow cytometry

Cells were fixed using 100 μl chilled 4% PFA in PBS (Thermo Fisher Scientific) at room temperature for 20 min and subsequently washed with Fluorescence-Activated Cell Sorting (FACS) buffer (1% BSA and 1% FBS in PBS, pH 7.2) to remove excess PFA. Cells were then plated in a V-bottom 96-well tissue culture plate (Cellstar) at $1 \times 10^6$ cells per well and washed by centrifugation at 1,000*g* for 3 min. Cells were resuspended in 100 μl FACS buffer supplemented with Fc-γ receptor anti-CD16/CD32 antibodies (1:100; Thermo Fisher Scientific) and incubated on ice at 4°C for 30 min, to ensure Fc receptors were blocked. Cells were then washed and incubated with 100 μl FACS buffer containing the relevant surface protein–targeted fluorophore-conjugated antibody (1:100, Table 3) on ice at 4°C for 30 min. Samples were resuspended in 400 μl FACS buffer and analysed using a BD FACSymphony A5 flow cytometer (BD Biosciences). Gating and data analysis were performed with either unstained samples or fluorescence minus

one control using FlowJo v10.8.1 (BD Biosciences), or FCS Express (De Novo Software).

## Data Availability

The mass spectrometry proteomic data have been deposited to the ProteomeXchange Consortium via the PRIDE partner repository with the dataset identifier: PXD051067.

## Supplementary Information

## Acknowledgements

This research was partly funded by a Wellcome Trust Investigator Award (215542/Z/19/Z) and a multi-user equipment grant (212947/Z/18/Z), a BBSRC CASE studentship to F Sidgwick, and an MRC DiMeN DTP Studentship to J Gudgeon.

### Author Contributions

J Gudgeon: formal analysis, investigation, methodology, and writing—original draft.
A Dannoura: investigation.
R Chatterjee: investigation.
F Sidgwick: investigation and methodology.
BBA Raymond: investigation.
AM Frey: formal analysis.
JL Marin-Rubio: supervision, investigation, and methodology.
M Trost: conceptualisation, supervision, funding acquisition, project administration, and writing—review and editing.

### Conflict of Interest Statement

The authors declare that they have no conflict of interest.

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
