## [Reviewer comments · Life Science Alliance]

Life Science Alliance

Mass spectrometry-based proteomic exploration of diverse murine macrophage cellular models

Jack Gudgeon, Abeer Dannoura, Ritika Chatterjee, Frances Sidgwick, Benjamin Raymond, Andrew Frey, Jose Luis Marin-Rubio, and Matthias Trost

DOI: <https://doi.org/10.26508/lsa.202402760>

Corresponding author(s): Matthias Trost, Newcastle University

Review Timeline:

Submission Date:	2024-04-08
Editorial Decision:	2024-05-10
Revision Received:	2024-10-01
Editorial Decision:	2024-10-17
Revision Received:	2024-10-18
Accepted:	2024-10-18

Transaction Report:

May 10, 2024

Re: Life Science Alliance manuscript #LSA-2024-02760

Prof. Matthias Trost
Newcastle University
ICAMB
Framlington Place
Newcastle-upon-Tyne NE24HH
United Kingdom

Dear Dr. Trost,

Thank you for submitting your manuscript entitled "Mass spectrometry-based proteomic exploration of diverse murine macrophage cellular models" to Life Science Alliance. The manuscript was assessed by expert reviewers, whose comments are appended to this letter. We invite you to submit a revised manuscript addressing the Reviewer comments.

Thank you for this interesting contribution to Life Science Alliance. We are looking forward to receiving your revised manuscript.

Sincerely,

B. MANUSCRIPT ORGANIZATION AND FORMATTING:

Reviewer #1 (Comments to the Authors (Required)):

Before supplying specific comments, it is important to point out that the Heap et al. paper is very important in the field of macrophage biology as it showed substantial population of L929 conditioned media (as a source of CSF-1) with soluble proteins that can have a multitude of direct and indirect effects on macrophage function and survival. Heap et al. substantiated the concept of favoring the use of recombinant CSF-1 as the differentiation agent, simply for reproducibility (e.g., Murray et al. 2014). L929 supernatants vary widely from lab-to-lab, in addition to the fact that few investigators assay how much CSF-1 is present in different batches. Accordingly, while L929 media is frequently viewed as a "cost effective alternative" to CSF-1, this is actually a classic case of false economy given that reproducibility and artifacts abound as a consequence of uncontrolled L929 generation and deployment.

The same group now advocates an even worse approach: use of transformed "macrophages" as a "cost effective" approach, or to satisfy a 3R imperative. Unfortunately, neither the data nor the hypothetical "savings" withstand scrutiny as a viable, practical approach.

Some comments include:

1. The authors use L929 supernatant as the CSF-1 source throughout even though Heap et al. showed this approach will lessen reproducibility and create artifacts that cannot be controlled.
2. The use of cell lines is another false economy situation. The authors own data show vast differences in the proteomes in each cell type, independence from CSF-1, and activation of massive changes (inflammation, key metabolic changes) linked to sustenance in in vitro culture. The end result of this approach is both a practical and hypothetical mess: should a new manuscript or finding be reported with the approach advocated by Gudgeon, a reviewer/editor/interested scientist will demand/performance the experiments with primary BMDMs. How does this satisfy a 3R requirement if everything ends up being repeated by many labs with BMDMs?
3. As a practical point, RAW264.7 macrophages must be cultivated under BSL2/S2 conditions in Germany. This raises many practical issues as the use of these cells requires a specialized S2 lab and substantial governmental bureaucratic measures.
4. Throughout their comparative proteomic approach, massive changes in the proteome are shown. It is unclear how the authors can conclude J774.1 cells are "functionally similar" to BMDMs when (i) no functional studies were performed and (ii) ~2500 proteins - that over 10% of the proteome at minimal, are different between the two cells.
5. The authors should recognize that any experienced macrophage-centric lab will (i) reduce their use of mice by freezing BM for future use, especially from KO and Tg mice without having to "re-start" a colony (including sharing with collaborators), (ii) will reduce their mouse use and increase reproducibility by using highly purified, LPS -ve, recombinant human CSF-1, and avoid the problems associated with cell line systems at all costs.

Reviewer #2 (Comments to the Authors (Required)):

The study of provides very interesting examination of the common macrophage cell lines. It lays out various processes that may differ in the cell lines compared to primary cells and how it may affect different macrophage studies. Through this, the paper really emphasizes that careful consideration should be taken when selecting a macrophage cell line or whether to use primary macrophages. Further, it shows that making your own immortalized BMDM cell line may not give the expected result either. The data as presented give an insight into what might change between the different lines, but more biological relevant experiments might support the claims further.

Question 1: Taking account MSR1 being a phagocytic receptor, could the authors provide phagocytic activity in different cell lines?

Question 2: The authors showed different TLRs-induced response in the different cell lines, could the authors support this claim

by stimulation of different cell lines with (ex. LPS) to see the difference in responses?

Question 3: Regarding table 1, It would be interesting to see the information about DNA/RNA sensors as well.

Question 4: It's also mentioned that two of the cell lines are derived from Balb/c mice. It would be interesting to see how BMDMs from Balb/c mice compare to C57BL/6J BMDMs.

Minor point: Page 4. Line: 109: The authors mentioned that WT and MSR1 KO were generated from C57BL/6N. In methods, Page 29, line 593 and 598, the authors mentioned C57bl/6J or C57BL/6, respectively.

Reviewer #3 (Comments to the Authors (Required)):

The manuscript by Gudgeon et al. presents a series of experiments that characterized the proteome of commonly-used macrophage cell-lines, compared to the standard MCSF-derived BMDM and newly-created J2-derived immortal lines derived from WT and Msr1-deficient mice.

As macrophages are crucial players in several human diseases, the characterization of faithful relevant models of rodent origin are of high interest. In addition, given the already vast collection of mouse transgenic and knockouts available from public repositories, that the validation of macrophage models with the J2-retrovirus system is an important task for the macrophage field. Finally, but very important, as stated by the authors, the reduction in the numbers of laboratory mice is also a priority in biomedicine. In those senses, the study is pertinent and has interest for the macrophage and immunity fields.

Authors are experts in proteomic analysis, and therefore, the overall insight in proteome knowledge is great. The effort in comparing proteomes from all 4 different lines, and the primary cells is valuable. However, I would recommend some suggestions that may improve the overall scientific impact of this study.

-As stated before, the overall proteomic depth of the study is prominent. However, there is a significant lack of functional assays. Most results are just confronted with literature. While the perspective of this work in the context of the already-published literature is certainly important, there are some experiments that could easily be added to improve the present work.

-For example: IFNGR1 result. As stated by the authors, this is surprising. Is this functionally relevant? While for example RAW cells certainly have been proven to respond to recombinant IFNg, is this result really unique to this setting? Do all 4 lines do not or do respond to IFNg? It would be easily tested for example by analyzing NO subproducts by iNOS expression. Do these cells respond to IFNg by expressing iNOS? Differentially expressed with BMDM?

-related to the previous. As a study like the present one has not been conducted, it would be informative to test the differential response of the lines and J2-derived iBMDM in response to LPS and IL-4 for example. As readout, LPS-dependent expression of Nos2, IL-1b or IL-6 contrasted with IL-4-dependent Arg1, FIZZ1, IL-10, or Retna. These would at least cover some of the favorite genes related to pro-inflammatory or anti-inflammatory profiles.

-The Msr1 expression, both at the protein and RNA level is also striking in iBMDMs. What is exactly the individual role of MSR1 in macrophages? As there are several, dozens of scavenger receptors, the degree of compensation would also be possible... However, the result is interesting: Is the expression of Msr1 in WT iBMDM similar to Msr1-KO iBMDM? Figure 1c should also be implemented with Msr1-KO iBMDM, to see if WT iBMDM express similar levels as KO iBMDM. This would favor the idea of an insertion. An alternative (although time-consuming), is to generate an independent biological replica of iBMDM line, to see if Msr1 is also down. A third option, that would be nice, is to check the nascent RNA. Measuring nascent hnRNA would allow to discern between promoter transcription or Msr1 mRNA stability. HnRNA is a surrogate for newly transcribed mRNA. It can be quantified through qPCR using primer pairs, with one primer located internally to an exon and the other spanning the exon/intron boundary. If newly RNA is transcribed in iBMDM, then other mechanisms besides insertion would be possibly acting.

-One additional important aspect is the mouse strain origin. As authors have noted in the discussion, the C57 of BALBc origin would make an impact. Regarding this, the pioneer work from Chris Glass and colleagues regarding macrophage phenotypes depending on the mouse strain should be mentioned and discussed possible implications.

-One last recommendation: Please try to banish the terms of M1 and M2. As these terms were useful one or two decades ago, and represent extreme states of polarization, there are plenty of immune and macrophage associations that recommend the change to "pro-inflammatory activation" for M1 or "anti-inflammatory" or tolerogenic for M2.

We would like to thank all reviewers for their comments which improved this paper.

Reviewer #1 (Comments to the Authors (Required)):

Before supplying specific comments, it is important to point out that the Heap et al. paper is very important in the field of macrophage biology as it showed substantial population of L929 conditioned media (as a source of CSF-1) with soluble proteins that can have a multitude of direct and indirect effects on macrophage function and survival. Heap et al. substantiated the concept of favoring the use of recombinant CSF-1 as the differentiation agent, simply for reproducibility (e.g., Murray et al. 2014). L929 superantants vary widely from lab-to-lab, in addition to the fact that few investigators assay how much CSF-1 is present in different batches. Accordingly, while L929 media is frequently viewed as a "cost effective alternative" to CSF-1, this is actually a classic case of false economy given that reproducibility and artifacts abound as a consequence of uncontrolled L929 generation and deployment.

The same group now advocates an even worse approach: use of transformed "macrophages" as a "cost effective" approach, or to satisfy a 3R imperative. Unfortunately, neither the data nor the hypothetical "savings" withstand scrutiny as a viable, practical approach.

We thank the reviewer who agrees that our previous paper is very important. Indeed, it has been cited 55 times since 2021 (Google scholar).

However, we believe that this reviewer does not fully understand our intentions: we want to compare commonly used tools in the macrophage field. Since many countries have strong restrictions on mouse work, we still believe that these models have value. While these cell models clearly have no relevance in classical immunology, they are important, specifically in biochemical work that requires sometimes tens of millions of cells per replicate.

In 2023 alone, 1397 Pubmed articles had the keyword RAW264.7, showing that cell line models of macrophages are still being used intensively. This paper will help researchers to understand the deficits of working with cell lines and their suitability for specific experiments.

Some comments include:

1. The authors use L929 supernatant as the CSF-1 source throughout even though Heap et al. showed this approach will lessen reproducibility and create artifacts that cannot be controlled.

Considering that this is our own paper, we do not share the opinion that this is one of the main outcomes of this paper. We found indeed that there are some differences between M-CSF and L929 medium but that these are not as big as some may think. As for reproducibility: yes, between labs and between days of L929 supernatant harvest, there are differences, but once the

supernatant is combined, within this, the data is reproducible. We assume that using m-CSF from different suppliers will also induces differences.

Furthermore, BMDMs, which are considered the gold-standard, are already an artificial model. Using a single growth/differentiation factor such as m-CSF is entirely artificial and does not represent a physiological situation either. As we have done in most of our papers in the past, we do believe that initial biochemical work in cell lines or BMDMs is justified, although we agree that physiological relevance needs to be tested in *in vivo* models whenever possible.

2. The use of cell lines is another false economy situation. The authors own data show vast differences in the proteomes in each cell type, independence from CSF-1, and activation of massive changes (inflammation, key metabolic changes) linked to sustenance in *in vitro* culture. The end result of this approach is both a practical and hypothetical mess: should a new manuscript or finding be reported with the approach advocated by Gudgeon, a reviewer/editor/interested scientist will demand/perform the experiments with primary BMDMs. How does this satisfy a 3R requirement if everything ends up being repeated by many labs with BMDMs?

Yes, results are often validated using primary BMDMs. However, in cases where experimental methods or assays need developing/optimising, the use of primary cells is not always practical. Further to this, genetic manipulation of primary BMDMs, required for experiments such as tag-based co-immunoprecipitation, is challenging, and requires large numbers of cells. Therefore, the use of cell lines for these types of experiments is not a false economy situation and is useful in satisfying 3R requirements.

3. As a practical point, RAW264.7 macrophages must be cultivated under BSL2/S2 conditions in Germany. This raises many practical issues as the use of these cells requires a specialized S2 lab and substantial governmental bureaucratic measures.

We were not aware of this fact, but we believe it does not affect our work.

4. Throughout their comparative proteomic approach, massive changes in the proteome are shown. It is unclear how the authors can conclude J774.1 cells are "functionally similar" to BMDMs when (i) no functional studies were performed and (ii) ~2500 proteins - that over 10% of the proteome at minimal, are different between the two cells.

We have changed the wording to "J774A.1 cells hold the most similar proteome compared to WT BMDMs".

5. The authors should recognize that any experienced macrophage-centric lab will (i) reduce their use of mice by freezing BM for future use, especially from KO and Tg mice without having to "re-start" a colony (including sharing with collaborators, (ii) will reduce their mouse use and increase reproducibility by using highly purified, LPS -ve, recombinant human CSF-1, and avoid the problems associated with cell line systems at all costs.

We agree that this is one possibility, and we have done this in the past. We have seen, however,

● / ● = Log₂FC of (+/-) 0.5-2 ●● / ●● = Log₂FC of (+/-) 2-4 ●●● / ●●● = Log₂FC of (+/-) >4
● = not detected NA = <3 valid values NS = No significant change

that freezing bone marrow for future use drastically changes the proteome and receptor expression of BMDMs, therefore this decreases reproducibility. Furthermore, cell number is drastically reduced using frozen bone marrow, meaning a larger number of mice are needed to satisfy experimental needs.

Reviewer #2 (Comments to the Authors (Required)):

The study of provides very interesting examination of the common macrophage cell lines. It lays out various processes that may differ in the cell lines compared to primary cells and how it may affect different macrophage studies. Through this, the paper really emphasizes that careful consideration should be taken when selecting a macrophage cell line or whether to use primary macrophages. Further, it shows that making your own immortalized BMDM cell line may not give the expected result either. The data as presented give an insight into what might change between the different lines, but more biological relevant experiments might support the claims further.

Question 1: Taking account MSR1 being a phagocytic receptor, could the authors provide phagocytic activity in different cell lines?

We have indeed now performed a phagocytosis assay showing that loss of MSR1 – as expected – influences uptake negatively charged beads, but not all particles tested. We added this as an additional Figure 9.

Question 2: The authors showed different TLRs-induced response in the different cell lines, could the authors support this claim by stimulation of different cell lines with (ex. LPS) to see the difference in responses?

We have performed multiple qPCR experiments in response to IL-4, LPS and IFN γ . This can be found in Figure 10.

Question 3: Regarding table 1, It would be interesting to see the information about DNA/RNA sensors as well.

We have appended Table 1 with the DNA and RNA sensors included below.

Gene name	Function	J774	RAW	BMA	iBMDM
DNA/RNA sensors					
Adar	Edits both viral and cellular dsRNA. Can have proviral or antiviral effects.	●	●	●	●
Aim2	Mediates inflammasome activation in response to dsDNA.	NA	NA	NA	NA
Cgas	Senses dsDNA, triggering type-I IFN production.	●●●	●●	●●	●●
Ddx58	Senses dsRNA, triggering type-I IFN production.	NS	●	●	NS
Mavs		NS	NS	NS	●
Ifih1	Senses dsRNA, triggering type-I IFN production.	NS	NS	●●	NA
Sting1	Senses dsDNA, triggering type-I IFN production.	●	NS	●●	●
TLR3	Senses dsRNA, triggering type-I IFN production.	NA	●	●	●
TLR9	Recognises unmethylated CpG dinucleotides.	NS	NS	NA	NS
ZBP1	Recognises Z-RNA structures, mediating pyroptosis, necroptosis and apoptosis.	NA	NA	●	●

Question 4: It's also mentioned that two of the cell lines are derived from Balb/c mice. It would be interesting to see how BMDMs from Balb/c mice compare to C57BL/6J BMDMs.

This is a good point. We have added a comparison of BMDMs from C57BL/6 vs BALB/c mice in the manuscript (Figure 8).

Minor point: Page 4. Line: 109: The authors mentioned that WT and MSR1 KO were generated from C57BL/6N. In methods, Page 29, line 593 and 598, the authors mentioned C57bl/6J or C57BL/6, respectively.

The manuscript has been corrected to include C57BL/6J mice as the source of BMDMs used in the study.

Reviewer #3 (Comments to the Authors (Required)):

The manuscript by Gudgeon et al. presents a series of experiments that characterized the proteome of commonly-used macrophage cell-lines, compared to the standard MCSF-derived BMDM and newly-created J2-derived immortal lines derived from WT and Msr1-deficient mice. As macrophages are crucial players in several human diseases, the characterization of faithful relevant models of rodent origin are of high interest. In addition, given the already vast collection of mouse transgenic and knockouts available from public repositories, that the validation of macrophage models with the J2-retrovirus system is an important task for the macrophage field.

Finally, but very important, as stated by the authors, the reduction in the numbers of laboratory mice is also a priority in biomedicine. In those senses, the study is pertinent and has interest for the macrophage and immunity fields.

Authors are experts in proteomic analysis, and therefore, the overall insight in proteome knowledge is great. The effort in comparing proteomes from all 4 different lines, and the primary cells is valuable. However, I would recommend some suggestions that may improve the overall scientific impact of this study.

As stated before, the overall proteomic depth of the study is prominent. However, there is a significant lack of functional assays. Most results are just confronted with literature. While the perspective of this work in the context of the already-published literature is certainly important, there are some experiments that could easily be added to improve the present work.

1. For example: IFNGR1 result. As stated by the authors, this is surprising. Is this functionally relevant? While for example RAW cells certainly have been proven to respond to recombinant IFNg, is this result really unique to this setting? Do all 4 lines do not or do respond to IFNg? It would be easily tested for example by analyzing NO subproducts by iNOS expression. Do these cells respond to IFNg by expressing iNOS? Differentially expressed with BMDM?

We have anticipated this request. As it turns out, the proteomics results were down to the fact that the few peptides of the protein were close to the detection limit and therefore not picked up in some runs. We have performed flow cytometry to show surface expression of the IFN-g receptor subunits and it shows only minor changes.

We have changed the text accordingly.

2. Related to the previous. As a study like the present one has not been conducted, it would be informative to test the differential response of the lines and J2-derived iBMDM in response to LPS and IL-4 for example. As readout, LPS-dependent expression of Nos2, IL-1b or IL-6 contrasted with IL-4-dependent Arg1, FIZZ1, IL-10, or Retna. These would at least cover some of the favorite genes related to pro-inflammatory or anti-inflammatory profiles.

We have performed multiple qPCR experiments in response to IL-4, LPS and IFNg. This can be found in Figure 10.

3. The Msr1 expression, both at the protein and RNA level is also striking in iBMDMs. What is exactly the individual role of MSR1 in macrophages? As there are several, dozens of scavenger receptors, the degree of compensation would also be possible... However, the result is interesting: Is the expression of Msr1 in WT iBMDM similar to Msr1-KO iBMDM? Figure 1c should also be implemented with Msr1-KO iBMDM, to see if WT iBMDM express similar levels as KO iBMDM. This would favor the idea of an insertion. An alternative (although time-consuming), is to generate an independent biological replica of iBMDM line, to see if Msr1 is also down. A third option, that would be nice, is to check the nascent RNA. Measuring nascent hnRNA would allow to discern between promoter transcription or Msr1 mRNA stability. HnRNA is a surrogate for newly transcribed mRNA. It can be quantified through qPCR

using primer pairs, with one primer located internally to an exon and the other spanning the exon/intron boundary. If newly RNA is transcribed in iBMDM, then other mechanisms besides insertion would be possibly acting.

We have included further mass spectrometry analysis of different iBMDM cell lines generated from C57BL6/Ntac BMDMs, in which MSR1 was also lost (Figure 1c). To clarify this further, we have included data below that shows the loss of MSR1 by data-dependent acquisition (DDA) mass spectrometry analysis in 4 biological replicates of iBMDM. This data has not been included or discussed in the manuscript as it was replaced by the more in-depth DIA analysis.

4. One additional important aspect is the mouse strain origin. As authors have noted in the discussion, the C57 of BALBc origin would make an impact. Regarding this, the pioneer work from Chris Glass and colleagues regarding macrophage phenotypes depending on the mouse strain should be mentioned and discussed possible implications.

As this was also requested by reviewer 2, we have now included a comparison of BMDMs of BALB/c with C57BL6 mice.

The study by Chris Glass has also been briefly included in the discussion.

5. One last recommendation: Please try to banish the terms of M1 and M2. As these terms were useful one or two decades ago, and represent extreme states of polarization, there are plenty of

immune and macrophage associations that recommend the change to "pro-inflammatory activation" for M1 or "anti-inflammatory" or tolerogenic for M2.

We agree with the reviewer. We have removed this and used the M(IL4)/M(IFNg) nomenclature.

October 17, 2024

RE: Life Science Alliance Manuscript #LSA-2024-02760R

Prof. Matthias Trost
Newcastle University
ICAMB
Framlington Place
Newcastle-upon-Tyne NE24HH
United Kingdom

Dear Dr. Trost,

Thank you for submitting your revised manuscript entitled "Mass spectrometry-based proteomic exploration of diverse murine macrophage cellular models". We would be happy to publish your paper in Life Science Alliance pending final revisions necessary to meet our formatting guidelines.

- please be sure that the authorship listing and order is correct
- please upload your main manuscript text as an editable doc file
- please upload your supplementary figures as single files
- please add the author contributions and a conflict of interest statement to the main manuscript text
- please use the [10 author names, et al.] format in your references (i.e. limit the author names to the first 10)
- please add a separate figure legend section to the main manuscript text
- please add a figure callout for Figure 3a to your main manuscript text

LSA now encourages authors to provide a 30-60 second video where the study is briefly explained. We will use these videos on social media to promote the published paper and the presenting author (for examples, see <https://docs.google.com/document/d/1-UWCfbE4pGcDdcgzcmiuJI2XMBJnxKYeqRvLLrLS08s/edit?usp=sharing>). Corresponding or first-authors are welcome to submit the video. Please submit only one video per manuscript. The video can be emailed to contact@life-science-alliance.org

A. FINAL FILES:

B. MANUSCRIPT ORGANIZATION AND FORMATTING:

Sincerely,

Reviewer #2 (Comments to the Authors (Required)):

The authors answered all my concerns and I recommend it for publication.

Reviewer #3 (Comments to the Authors (Required)):

Authors have answered the revision requests. I have no further queries.

October 18, 2024

RE: Life Science Alliance Manuscript #LSA-2024-02760RR

Prof. Matthias Trost
Newcastle University
ICAMB
Framlington Place
Newcastle-upon-Tyne NE24HH
United Kingdom

Dear Dr. Trost,

Thank you for submitting your Resource entitled "Mass spectrometry-based proteomic exploration of diverse murine macrophage cellular models". It is a pleasure to let you know that your manuscript is now accepted for publication in Life Science Alliance. Congratulations on this interesting work.

DISTRIBUTION OF MATERIALS:

Again, congratulations on a very nice paper. I hope you found the review process to be constructive and are pleased with how the manuscript was handled editorially. We look forward to future exciting submissions from your lab.

Sincerely,
